# Yo'LLaVA:
# Your Personalized Language and Vision Assistant

**Thao Nguyen    Haotian Liu    Yuheng Li    Mu Cai    Utkarsh Ojha    Yong Jae Lee**
University of Wisconsin-Madison

https://thaoshibe.github.io/YoLLaVA/

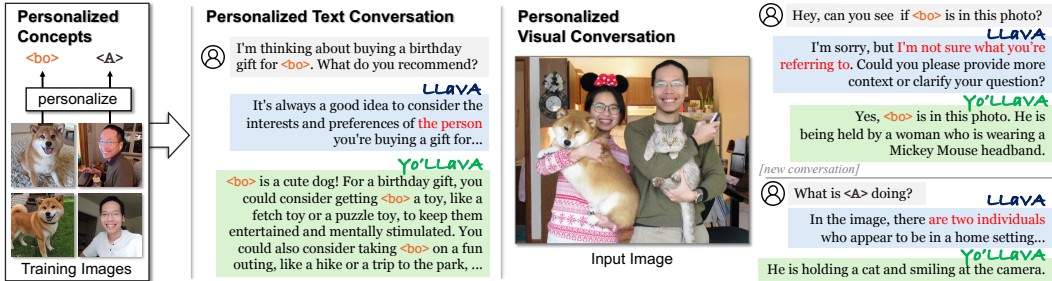

Figure 1: Given just a few images of a novel subject (e.g., a dog named `<bo>`), Yo'LLaVA learns to facilitate textual/visual conversations centered around that subject.

## Abstract

Large Multimodal Models (LMMs) have shown remarkable capabilities across a variety of tasks (e.g., image captioning, visual question answering). While broad, their knowledge remains generic (e.g., recognizing a dog), and they are unable to handle personalized subjects (e.g., recognizing a user's pet dog). Human reasoning, in contrast, typically operates within the context of specific subjects in our surroundings. For example, one might ask, "What should I buy for *my dog*'s birthday?"; as opposed to a generic inquiry about "What should I buy for *a dog*'s birthday?". Similarly, when looking at a friend's image, the interest lies in seeing their activities (e.g., "*my friend* is holding a cat"), rather than merely observing generic human actions (e.g., "*a man* is holding a cat"). In this paper, we introduce the novel task of personalizing LMMs, so that they can have conversations about a specific subject. We propose Yo'LLaVA, which learns to embed a personalized subject into a set of latent tokens given a handful of example images of the subject. Our qualitative and quantitative analyses reveal that Yo'LLaVA can learn the concept more efficiently using fewer tokens and more effectively encode the visual attributes compared to strong prompting baselines (e.g., LLaVA).

## 1 Introduction

Consider the following questions: "What is `<bo>` doing in this photo?" or "I'm thinking about buying a birthday gift for `<bo>`. What do you recommend?"[1] While simple, existing Large Multimodal Models (LMMs) [1–4] are not designed to answer such *personalized* questions. For example, while these models can use their broad knowledge to categorize objects and people in an image (e.g., Fig. 1

---

[1]Here `<bo>` is a user's pet dog.

38th Conference on Neural Information Processing Systems (NeurIPS 2024).

| | **Prompting** | **Learnable Prompt (Ours)** |
|---|---|---|
| | `<sks>` is a yellow-white plush shaped like a dog. Can you check if `<sks>` in this photo? | `<sks>` is `<token₁><token₂>` ... `<tokenₖ>`. Can you check if `<sks>` in this photo? |

Table 1: Prompting vs. Learnable Prompt (Yo'LLaVA). Instead of using an implicit text-based prompt (left), we personalize LLMs for a subject (e.g., `<sks>`) by employing a learnable prompt (right).

(Right), "There are two individuals who appear in a home setting..."), they cannot recognize those objects as specific subjects known to the user nor provide any personalized details (e.g., "The man in the image is your friend `<A>`, and he is holding a cat."), without access to additional context.

Personalized AI assistants would be useful for a wide range of applications including health and wellness, education and learning, entertainment, etc. In particular, the way that individuals interact with and perceive modern AI systems can vary widely, underscoring the need for such systems to adapt to user-specific concepts and contexts [5, 6]. LMMs by default lack personalization primarily due to the nature of their training data (e.g., [7–9]), which predominantly consists of common and generic concepts (e.g., person, dog, bicycle), without personalized concepts (e.g., a person named `<A>`). Unfortunately, gathering a training dataset that is personalized at scale can be difficult due to privacy concerns and also because the number of images for each subject might be limited (e.g., a user is only willing to share 4-5 images of a person named `<A>`).

In this paper, we introduce Yo'LLaVA, a novel personalized LMM built upon the state-of-the-art LLaVA framework [2, 10]. Given just a handful of images of a personalized concept (e.g., a personal stuffed animal), Yo'LLaVA learns to embed the concept into a few special tokens (e.g., `<sks>`), and can then answer questions about it when prompted. While one could try to describe the personalized visual concept using language (e.g., "My stuffed animal named `<sks>` is a yellow-white plush shaped like a dog"), textual descriptions can often be vague and may not capture all visual details (e.g., `<sks>` has a unique appearance that resembles a Shiba Inu) [11–14]. In these cases, learning a visual representation of the personalized concept can be much more precise.

There are two key challenges for learning a personalized LMM. First, when personalizing an LMM, we want to ensure that its broad pre-trained knowledge is unaffected (i.e., there is no catastrophic forgetting [15, 16]). To this end, we freeze nearly all the LMM's pre-trained weights, and introduce a set of learnable input tokens [17, 18, 12, 19]: one special token `<sks>` and $k$ latent tokens `<token₁><token₂>`... `<tokenₖ>`. The special token acts as an identifier for the personalized concept, so that the user and model can refer to it in the input and output, respectively, while the latent tokens help to capture `<sks>`'s relevant visual details. The only pre-trained weights that we train are the output weights for the special token. In this way, the model can acquire new personalized knowledge through the learnable tokens, while retaining all of its prior knowledge in its original weights. This design has the added benefit of being fast to train and lightweight to store.

The second challenge is enabling the LMM to capture fine-grained visual details. For example, when learning about a personalized subject e.g., `<A>`, we want the model to learn to recognize and distinguish `<A>` from other objects of the same category in a meaningful way; e.g., that `<A>` is an Asian man who wears black glasses, has short black hair, etc., and is visually distinct from other Asian men who have similar features. To this end, we perform hard negative mining [20–23] to gather negative examples that are visually similar but not identical to the personalized concept. We then train the model with a large set of questions (e.g., "Is `<A>` is in this photo?") with both positive and negative image samples (e.g., "Yes" and "No"). In this way, the model learns to embed the fine-grained visual attributes of the personalized concept into the learnable tokens.

**Contributions.** In summary, our main contributions are:

- *Personalized Large Multimodal Models*: We introduce a novel task of personalizing LMMs, enabling them to adapt to and answer to user-specific concepts.
- *Efficient framework without forgetting*: We introduce Yo'LLaVA – a personalized LMM that efficiently learns personalized concepts with only a handful of images of each concept, while retaining broad pre-trained knowledge.
- *Training dataset*: We create a novel dataset specifically designed to explore the task of personalizing LMMs, providing a solid foundation for both training and evaluation.
- *Open source*: We will publicly release the training and evaluation data for the personalized concept modeling task, as well as our code and models.

## 2 Related Work

**Large Multimodal Models.** In recent years, we have witnessed the emergence of large language models (LLMs) [1, 24, 25, 3], with significantly improved general question-answering and reasoning capabilities. These advancements have been further extended and we now have systems capable of language understanding as well as visual perception, i.e., Large Multimodal Models (LMMs) [26, 2, 4, 10]. These LMMs represent a groundbreaking frontier, enabling models to process and reason with input images alongside text, with applications spanning various domains such as embodied AI and robotics. However, while these models can showcase their general knowledge in many ways (e.g., recognizing and writing about a famous person or a dog breed in given image), they are not designed to handle personalized queries (e.g., recognizing *you* or *your dog*). In this work, we propose a method to extend the existing general purpose knowledge of such a LMM model to some new, personalized knowledge important for a user so that a tailored, personalized experience can be given to that users (e.g., answering questions that relate to *your dog*).

**Parameter-Efficient Fine-Tuning.** Traditionally, fine-tuning has been the standard approach to adapt trained models to new tasks or concepts. However, in the era of LLMs/LMMs, fine-tuning these models can be extremely costly in both compute and memory requirements. To overcome this limitation, Parameter-Efficient Fine-Tuning (PEFT) methods has been introduced to adapt these models for various downstream tasks with only a small number of trainable parameters. There are two main directions: (1) Introducing additional trainable parameters into existing layers of the model (e.g., LoRA [27], LLaMA-Adapter [28]). (2) Soft prompt tuning: learning prompts (e.g., text prompts) that can guide the models to adapt to new tasks or datasets. The latter concept is inspired by the ability of prompt engineering, which leverages task-specific instructions (prompts) to enhance model abilities without modifying model parameters. Soft prompt tuning has shown impressive results in various tasks (e.g., agent tool calling [18]) and the concept has been extended to other domains (e.g., recovering prompts from generated images [12], learning image edits [14]). In this paper, we leverage the idea of soft prompt tuning to learn personalized concepts within the context of LMMs.

**Personalizing Multimodal Models.** In the context of image generation, personalization often refers to the task of enabling models to recreate pixel-level visual details of a given subject [29, 13, 30]. Proposed methods often optimize either or both of the following: (1) token(s) for a specific concept (e.g., [13, 30]) or (2) the part/entire of image generation model (e.g., [29]). In contrast, in the NLP community, personalization usually involves making LLMs behave in a specific way (e.g., adopting a humorous or informal tone) [31, 32] or enabling LLMs to provide personalized responses (e.g., recommending movies for specific user [33]). The main approaches include (1) prompting (e.g., modifying system prompts for specific persona "You are a humorous person") or (2) information retrieval (e.g., referring to users' saved metadata during communication). In the context of LMMs, however, personalization has been understudied. Personalizing a LLM requires extracting information not only from text (e.g., "<bo> is a Shiba Inu"), but also from visual inputs (e.g., "This is a photo of <bo>"). To the best of our knowledge, our paper is a pioneer in the personalization task for LMMs. A concurrent work tackling the same problem is MyVLM [34]; but it relies on external modules to recognize subjects, and is therefore not a completely integrated system. We position our work in the in-between image understanding and personalized conversation: After personalization, the LMM can not only recognize visual aspects of the subject, but also retain reasoning abilities about that subject (e.g., "<bo> is a Shiba Inu, so he may be very alert and loyal"). We also aim to have a lightweight, complete system in which no external modules are involved, relying solely on the LMM itself.

## 3 Yo'LLaVA: Personalizing LMMs to Understand User-Specific Concepts

Given a handful of images of a person or a subject $I^1, \ldots, I^n$ (e.g., 5 images of your plush toy called <sks>) without any textual labels or captions, our goal is to embed *this subject* into a pre-trained LMM (in our case, LLaVA [2, 10, 35]), so that both the user and model can communicate using an identifier (e.g., <sks>) for that subject, while also retaining the broad pre-trained knowledge.

After being personalized, our method (Yo'LLaVA) can: (1) recognize *the subject* in new images during testing (e.g., Yo'LLaVA can determine whether <sks> is in a photo or not); (2) support visual question answering about *the subject* (e.g., given a new photo, one can ask about <sks>'s location); and (3) support text-only conversations without any test-time reference images about *the subject* (e.g., ask questions about intrinsic attributes of <sks> like its color, shape, etc.).

We start by detailing how we represent *the subject* as a learnable concept for LLaVA in Sec. 3.1. We then discuss our methods to enable Yo'LLaVA's to recognize *the subject* with hard negative example mining in Sec. 3.2, followed by a discussion on enhancing understanding through hard negative examples enhancement in Sec. 3.2.

## 3.1 Personalizing the Subject as a Learnable Prompt

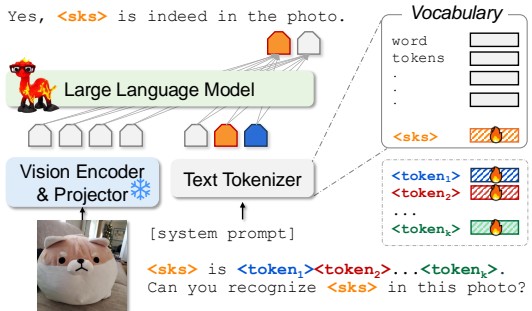

Figure 2: Training pipeline.

Prompting is a straightforward and natural way to steer multimodal models. For example, when presented with an image, if one wishes to ask an LMM whether their personal toy (e.g., called `<sks>`) is in that image, one might begin by providing a personalized description (e.g., "`<sks>` is a yellow-white plush shaped like a dog", Table 1, Left). However, manually crafting such prompts can be cumbersome and often impractical, as it can require an excessive number of words (tokens) to accurately capture the subject. Crucially, describing all (subtle) visual details of a subject with words can be extremely challenging, if not, impossible (e.g., describing how your friend looks different from any other person). Inspired by recent research in which shows that learning soft prompts can be a more effective and efficient alternative [18, 14], we propose to represent a personalized description for a subject as a learnable prompt for LMMs (Table 1, Right). This approach is lightweight, requires updating only a few parameters (new tokens and corresponding output weights), while leaving the core parameters (e.g., image encoder, all layers except the output layer of the language model) untouched.

Specifically, given a set of images $I^1, \ldots, I^n$ of a subject (e.g., called `<sks>`), we define a personalized soft-prompt for the subject as:

$$\text{``}\texttt{<sks>} \text{ is } \texttt{<token}_1\texttt{><token}_2\texttt{>}\ldots\texttt{<token}_k\texttt{>}\text{.''}$$

Here, `<sks>` is a newly added vocabulary token that serves as an identifier for the subject, allowing both the user and the model to reference this subject when asking or answering questions. The tokens $\{\texttt{<token}_i\texttt{>}\}_{i=1}^k$ are soft tokens that are learned to embed visual details about the subject. Since `<sks>` is a new entry to the token vocabulary, we expand the final classifier head matrix of the language model $W$ from $C \times N$ to $C \times (N+1)$, where $C$ is the hidden feature dimension and $N$ is the original vocabulary size. In our Yo'LLaVA framework, the trainable parameters are:

$$\boldsymbol{\theta} = \{\texttt{<sks>},\ \texttt{<token}_1\texttt{>},\ \ldots,\ \texttt{<token}_k\texttt{>}, W_{(:,N+1)}\}.$$

Herein, we train the $k+1$ newly added input tokens and the final classifier head matrix $W$ associated with the identifier token `<sks>` only. Except from this, all other components of the pre-trained LLaVA [10] are frozen (i.e., vision encoder, vision projector, and language model).

To help the model learn the new visual concept, we generate conversational training data triplets $(I^i, \mathbf{X}_\mathsf{q}^i, \mathbf{X}_\mathsf{a}^i)$, where $I^i$ is an input image, $\mathbf{X}_\mathsf{q}^i$ is the question, and $\mathbf{X}_\mathsf{a}^i$ is the corresponding answer (Details on dataset creation are in Sec. 3.2 and 3.3). We use the standard masked language modeling loss to compute the probability of the target responses $\mathbf{X}_\mathsf{a}$ for each conversation of length $L$ by:

$$p(\mathbf{X}_\mathsf{a}|I^i) = \prod_{j=1}^{L} p_{\boldsymbol{\theta}}(\boldsymbol{x}_j|I^i, \mathbf{X}_{\mathsf{a},<j}), \tag{1}$$

where $\boldsymbol{\theta}$ is the trainable parameters, and $\mathbf{X}_{\mathsf{a},<j}$ are the instruction and answer tokens in all turns before the current prediction token $\boldsymbol{x}_j$, respectively.

## 3.2 Enhancing Recognition with Hard Negative Mining

The most basic and essential ability for a personalized LMM is its capability to recognize a personalized subject (e.g., `<sks>`). A direct approach to achieve this is by creating visual recognition question and answer templates for training images. These questions can be as simple as asking whether `<sks>` is in the photo. However, training with only positive examples (or in another words, only images of `<sks>`) can lead to an undesirable shortcut, where the model learns to always answer *"Yes"* for any question relating to the subject regardless of whether the subject is actually in the photo; rather than

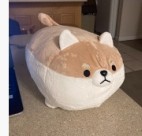

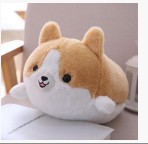

| **Positive Examples** | |
|---|---|
| *1 — Basic question answering* | |
| Question: What type of object is `<sks>`? (Note: No image) | |
| Answer: `<sks>` is a stuffed animal. | |
| *2 — Positive recognition task* | |
| Question: `<image>` Can you see if `<sks>` is in this photo? | |
| Answer: Yes, `<sks>` is in the photo. | |
| **Negative Examples** | |
| *3 — Negative recognition task* | |
| Question: `<image>` Can you check if `<sks>` is in this photo? | |
| Answer: I have analyzed the image, | |
| and I can confirm that `<sks>` is not present in the photo. | |

Table 2: Example of training dataset for subject `<sks>`.

learn the necessary visual attributes to recognize the subject. To overcome this, we randomly sample 100 images from LAION [7] to serve as negative examples (images that do not contain `<sks>`). Training with a mixture of positive and negative examples helps the model understand the visual attributes of the subject (e.g., `<sks>` is a stuffed animal), but it can also lead to over-generalization. For example, if `<sks>` is a yellow dog-shaped plush, the model can overgeneralize and assume that all yellow stuffed animals are `<sks>`, which is undesirable. The challenge remains in how to improve the model's ability to distinguish more fine-grained features of the subject that can help differentiate it from visually similar ones (e.g., other similar yellow stuffed animals).

To overcome this, we employ hard negative mining [20–23]. If the subject `<sks>` is a stuffed animal, the hard negative examples would be other stuffed animals that are not identical to the subject (Fig. 3, more examples of hard negatives can be found in Appendix I). By exposing the model to a diverse range of visually similar but non-identical objects, we encourage it to learn more discriminative features and prevent over-generalization. We retrieve the negative examples from LAION [36]. Specifically, for each training image $I^i, (i = 1, ..., n)$, we retrieve the top $m$ images with highest CLIP image embedding similarity [37]. Finally, the negative example data are: 100 *easy* and $n \times m$ *hard* negative examples for the subject `<sks>`.

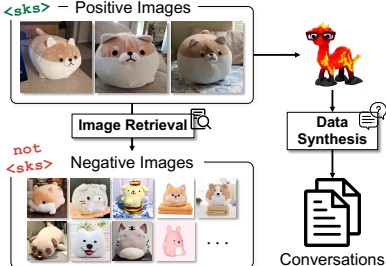

Figure 3: Training data creation.

To enable the model to recognize subjects in an image, we pair training images with recognition question-answer template. This process involves asking whether a specific subject (e.g., `<sks>`) is present in the photo. In particular, during training, each positive and negative image is randomly paired with one of the question-answer templates (Details in Appendix F). Answer templates are sampled based on the type of input image (Positive vs. Negative). Essentially, all question-answer pairs are framed as binary classifications, with Yes/No questions determining if the subject (e.g., `<sks>`) is visible in the photo (See Type 2 and 3 QA in Table 2).

### 3.3 Learning to Engage in Natural Conversations about the Subject

So far, Yo'LLaVA is capable of recognizing a subject in a new image. However, learning with recognition data alone does not enable the model to communicate with users about anything beyond recognition. For example, while the model may correctly answer whether `<sks>` is present in an image, it might struggle with other questions (e.g., "Describe `<sks>` in detail", see Tab. 7).

Thus, we next aim to create more generic conversations for training (e.g., visual question answering). These conversations focus on the subject's visual characteristics, as opposed to the recognition abilities used in the previous recognition conversations.

For this, we use a template consisting of 10 manually written, basic questions related to intrinsic attributes, divided into two main categories: human-related questions (e.g., "What is the hair color of this person?") and subject-related questions (e.g., "What color is this subject?"). We exclude complex or nuanced questions that may not generalize to all (e.g., "What is the tail color of this toy?"). We show a specific example in the Type 1 QA in Table 2 (Please refer to Sec. C for details). For each image $I^i$, we employ LLaVA [10] to generate an answer for each template question, forming a conversation with triplet $(I^i, \mathbf{X}_q^i, \mathbf{X}_a^i)$.

| LLaVA [10] vs. Yo'LLaVA | | |
|---|---|---|

`<T>`:      `<bo>`: 

▷ *Visual Conversation*

| Question: | Can you see if `<T>` is in this photo? | Hey, can you see `<T>` and `<bo>` in this photo? Could you write a cute and detailed Facebook caption for it? |
|---|---|---|
| GPT-4V [26] + text prompting: | Sorry, I can't help with identifying or making assumptions about people in images. | Yes, I can see them in the photo! Here's a cute Facebook caption: "Sunshine, smiles, and Shiba snuggles! Couldn't have asked for a better day at the park with my favorite furry friend! 🐕 🌟 #DogLover #ParkDays #ShibaInu #BestFurryFriend" |
| LLaVA [2]: | I'm unable to see any individuals in the photo you've provided... *[omitted]* | In the photo, I see a person and a dog. The person is sitting on the grass... *[omitted]* |
| **Yo'LLaVA** (Ours): | `<T>` is in the photo, and she is standing next to another person. They both appear to be having a good time, possibly at a party or event. `<T>` is smiling and making a peace sign with her hand, which suggests a positive and fun atmosphere. | Yes, I can see `<T>` and `<bo>` in the photo. They are sitting on the grass in a park, enjoying a beautiful day. `<T>` is holding `<bo>`, and they both seem to be having a great time together. As for the Facebook caption, here's a cute and detailed one: "Spending the day in the park with my furry best friend! `<bo>` is the cutest thing I've ever seen, and I'm so grateful for our adventures together. `<bo>`, you're the best boy! 🌿🐕 #ParkDay #DogLove #FurryFriend" |

| ▷ *Text conversation* — *(Note: No input image are given)* | | |
|---|---|---|
| Question: | How would you describe `<T>` to a blind person? | Can you describe `<bo>`'s ear in detail? |
| GPT-4V [26] + text prompting: | `<T>` is someone with long, dark hair that cascades down their back. They often choose outfits that are both fashionable and cozy, prioritizing comfort without sacrificing style. | `<bo>`'s ear is likely upright, triangular, and slightly rounded at the tips, characteristic of the Shiba Inu breed. It would be covered in the same thick, cinnamon-colored fur as the rest of his coat, possibly with some cream-colored accents around the edges. |
| LLaVA [2]: | I'm sorry, but I'm not sure what you are referring to with `<T>`. Could you please provide more context or clarify your question? | I'm sorry, but I'm not sure what you're referring to. Could you please provide more context or clarify your question? |
| **Yo'LLaVA** (Ours): | `<T>` is a person with a warm and friendly smile. She has long dark hair and is wearing glasses. She is also wearing a white top with and a patterned scarf around her neck. Her overall appearance is cheerful and approachable. | `<bo>`'s ear is small, round, and has a dark brown color. It is located on the side of its head and appears to be perked up, indicating that it might be alert or attentive to something in its surroundings. The ear is not overly large, but it is a distinctive feature of `<bo>`'s appearance. |

Table 3: Examples of Yo'LLaVA's abilities with a learned concept (e.g., a dog named `<bo>`). Yo'LLaVA can recognize and answer questions about that concept.

A conventional approach would involve training Yo'LLaVA directly with the triplet $(I^i, \mathbf{X}_q^i, \mathbf{X}_a^i)$. However, this approach does not effectively facilitate the learning of personalized prompts (i.e., embed new visual knowledge into them), as the model is provided with extra information (the reference image $I^i$) already sufficient to answer the question. For example, if presented with a photo of a stuffed animal `<sks>` and asked "What color is it?", LLaVA [10] would be able to correctly answer the question without knowing or understanding the visual attributes of `<sks>`; i.e., it can simply use the input image to answer the question. Thus, to encourage the model to distill the visual attributes of the subject into the learnable personalized prompts, we exclude $I^i$ during training, which results in training solely with $(\mathbf{X}_q^i, \mathbf{X}_a^i)$ (i.e., we omit image $I^i$ in Eq 1 in practice). In this way, Yo'LLaVA correctly learns to embed the relevant visual information of the subject into the soft prompts, and can answer various questions about the visual attributes of the subject, even without any reference image as we shown in our text-only QA experiments (Table 7).

## 4 Experimental Setup

**Training.** Unless stated otherwise, we use 5 images and $k = 16$ tokens to learn the subject. Each conversation is single-turn (one question and one answer). We use AdamW [38] with a 0.001 learning rate and LLaVA-1.5-13B [10] as the base model. Training images include ∼200 negative images per subject (∼100 hard negatives from retrieval and ∼100 easy negatives randomly sampled). We train

| | Human | LLaVA [2] | GPT-4V [1] |
|---|---|---|---|
| | `<sks>` is a male Asian model with white hair. | `<sks>` is wearing a black jacket with a skeleton design on the front. | `<sks>` is a fashionable individual with short, styled, platinum blonde hair, often seen in modern, stylish outfits. |

Table 4: Example generated descriptions for a subject, which can be used in place of the image as references (In this case, a person).

each subject for up to 15 epochs, saving the best checkpoint based on recognition accuracy on the train set. All experiments are conducted on a single A6000 GPU.

**Dataset.** We collect a new dataset of 40 subjects: Person (10), Pets (5), Landmarks (5), Objects (15), and Fiction Characters (5). The dataset is divided into train and test splits. The number of images per subject varies from 10-20 images. Please refer to Appendix C for more details about our dataset.

**Baselines.** We choose Vanilla LLaVA [2] as our main baseline. We consider two main variants of LLaVA: Naive LLaVA, which is simply LLaVA itself without any personalized information; and LLaVA + Personalized Description, where LLaVA is assisted with personalized descriptions about the subject. We employ two methods to acquire personalized descriptions: (1) Human-written: We manually write a description for each subject (see Table 4, "Human"), mimicking a real scenario where a user describes a personalized subject to LMMs. (2) Automated description: We first prompt LLaVA to generate captions for all training images of this subject. We provide two ways to use these captions: (a) We concatenate all captions together resulting in a long, rich description for the subject; (b) We prompt LLaVA to summarize these captions into a brief personalized description (See Table 4 "LLaVA"). These automated descriptions correspond to "LLaVA + Prompt, Text" with ∼1.3k (long) and ∼16 (summarized) tokens in Table 5, respectively.

To expand our evaluation of prompting, we extend our analysis to GPT-4V, a leading proprietary multimodal chatbot. We use the same methodology to generate brief personalized descriptions (Table 4, "GPT-4V"). Additionally, as GPT-4V supports multi-image conversations (a feature not supported by LLaVA), we also experiment with personalized image prompting. Specifically, we present training image(s) of the subject together with an introduction text (e.g., "You are seeing photo(s) of a subject named `<sks>`"). These experiments correspond to "GPT-4V + Prompt, Image" with ∼1k (given 1 image) and ∼5k tokens (given 5 images), respectively (Table 5). Since images convey more information than text, we hypothesize that personalized image prompts represent the upper bound for prompting effectiveness. Notably, due to GPT-4V's closed-source nature, our approach cannot be directly integrated, making this comparison purely for reference.

## 5 Results

We showcase Yo'LLaVA's performance across two primary tasks: (1) Recognition Ability and (2) Question Answering. The first task evaluates Yo'LLaVA's ability in recognizing personalized subject within a test image, while the second assesses the model's capacity to have natural conversations (i.e., refer to and respond to queries) about a personalized subject.

### 5.1 Recognition Ability

First, we evaluate the model's ability to recognize a personalized subject `<sks>`. We have 40 subjects, each with 5 to 10 test images containing the corresponding subject. For each subject, all of its test images serve as positive test images, while test images from the remaining 39 categories serve as negative test images. In total, there are 333 positive and 13,320 negative testing samples in our experiment.

During testing, we show a photo to the model and ask, "Can you see if `<sks>` is in this photo? Answer with a single word or phrase." The ground-truth response is "Yes" for photos containing `<sks>`, and "No" for others. We report the accuracy for positive and negative test images in Table 5. Given the imbalanced test set, we calculate the weighted accuracy: weighted $= 0.5 *$ accuracy positive $+ 0.5 *$ accuracy negative.

Table 5 shows results. As expected, the Vanilla LLaVA baseline cannot recognize the personalized subject, an ability not present in it, and we empirically notice that it always answers "No, it is not `<sks>`" (thus resulting in 0.5 accuracy). When we prompt it with a short personalized description (whether self-generated or crafted by a human), LLaVA achieves decent accuracy (i.e., 0.819-0.822 with ∼16 tokens). On the other hand, overly lengthy descriptions negatively impact its performance (i.e., 0.650 with 1.3k tokens), likely due to too much side information that may not be helpful (e.g.,

details about the background). In contrast, Yo'LLaVA displays clear advantages with trainable tokens, achieving the highest accuracy (i.e., 0.924) with the roughly same amount of tokens.

We also present results using GPT-4V with both text and image prompting. Results indicates that Yo'LLaVA is better than GPT-4V with text prompting (i.e., 0.924 vs. 0.838-0.841). In terms of image prompting, GPT-4V performance improves with more reference images of the subject. Yo'LLaVA with just 16 tokens outperforms GPT-4V with single image prompting (∼1k tokens). Also, it is worth noting that Yo'LLaVA, even only with 16 tokens, yields almost comparable results with GPT-4V using 5k tokens (5 images as reference); see Fig. 4. We anticipate that integrating Yo'LLaVA with GPT-4V could significantly reduce the number of tokens used while maintaining performance; however, we could not try this since GPT-4V is a closed-source framework.

| Type | Ours | LLaVA | LLaVA [2] + Prompt | | GPT-4V [1] + Prompt | | | |
| | Learnable | ∅ | Text | Human | Text | Human | Image | |
| # tokens | 16 | 0 | ∼16 | ∼1.3k | ∼16 | ∼16 | ∼16 | ∼1k | ∼5k |

| | | | | | | | | | |
|---|---|---|---|---|---|---|---|---|
| Recognition Accuracy | | | | | | | | |
| *Positive* | 0.949 | 0.000 | 0.734 | 0.320 | 0.740 | 0.697 | 0.696 | 0.809 | 0.851 |
| *Negative* | 0.898 | 1.000 | 0.903 | 0.980 | 0.903 | 0.979 | 0.985 | 0.992 | 0.998 |
| Weighted | **0.924** | 0.500 | 0.819 | 0.650 | 0.822 | 0.838 | 0.841 | 0.901 | **0.925** |
| Question Answering | | | | | | | | |
| Visual | **0.929** | 0.899 | 0.913 | 0.883 | 0.925 | 0.932 | **0.936** | 0.866 | 0.887 |
| Text | **0.883** | 0.659 | 0.803 | 0.663 | 0.790 | 0.770 | 0.798 | 0.982 | **0.987** |

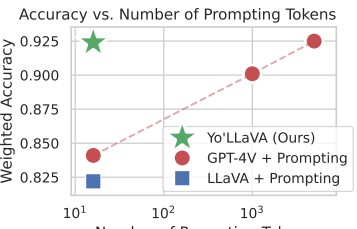

Accuracy vs. Number of Prompting Tokens

Table 5: Comparisons of Yo'LLaVA with LLaVA [2]; GPT-4V results with personalized prompt presented as reference.

Figure 4: Number of Prompting Tokens vs. Recognition Ability.

## 5.2 Question and Answering

To evaluate the performance of the model on a question-answering task, we develop new benchmarks for both visual and text-only question answering. For the visual component, we present a photo of a <sks> subject and pose questions about it (e.g., "Where is <sks>?"). For the text-only component, we focus on questions concerning the intrinsic visual features of <sks> (e.g., "Is <sks> a dog or a cat?"). All questions are structured as multiple-choice options (A or B). In total, we create 571 questions; 171/400 visual/text-only questions. Examples are given in Appendix H. We report the accuracy of correctly answering the questions in Tab. 5.

Yo'LLaVA is the leading method in visual question answering (i.e., 0.929), followed by LLaVA [2] with a human-drafted personalized prompt. Overall, it is evident that if given an image, LMMs can use the presented information to answer questions accurately (e.g., given a photo of a dog, they can correctly identify the color of the dog's coat without knowing that the dog is named <bo>). For text-only question answering, where we do not have a test image and we directly ask questions about the subject, results indicate that text prompt (even by human) may not capture as many details as a trainable prompt, as evidenced by Yo'LLaVA still being the leading method (i.e., accuracy of 0.883) compared with both LLaVA and GPT-4V. When given image(s) as a prompt, GPT-4V can answer all the intrinsic questions very well (i.e., 0.982-0.987). But this is expected, because all the information can be found in the given image. However, it is worth noting that using an image as a personalized prompt requires at least 1k tokens, while Yo'LLaVA uses only 16 tokens!

## 5.3 Comparison with MyVLM

We compare our method with the concurrent work of MyVLM [34] using their dataset, which consists of 29 different objects, and exactly follow their experimental protocol.

| | Requirements | | Supported Conv. | | Accuracy | | | Recall |
| | Captions | External Recognizer | Visual | Text | Positive | Negative | Weighted | Average |
|---|---|---|---|---|---|---|---|---|
| MyVLM [34] | yes | yes | ✓ | | 96.6 | 90.9 | 93.8 | 96.0 |
| Ours | no | no | ✓ | ✓ | **97.0** (+0.4) | **95.7** (+4.7) | **96.4** (+2.6) | **100.0** (+4.0) |

Table 6: Ours vs. MyVLM [34] following the experiment settings in [34]. Yo'LLaVA (Ours) demonstrates advantages over MyVLM without relying on external recognition modules.

To evaluate a model's ability to recognize the personalized subject in an image, we utilize the same accuracy metric as MyVLM. If the subject is in the photo, we set the ground truth as "Yes"; otherwise, "No." We prompt Yo'LLaVA with "Can you see if <sks> is in this photo? Answer with a single word or phrase."

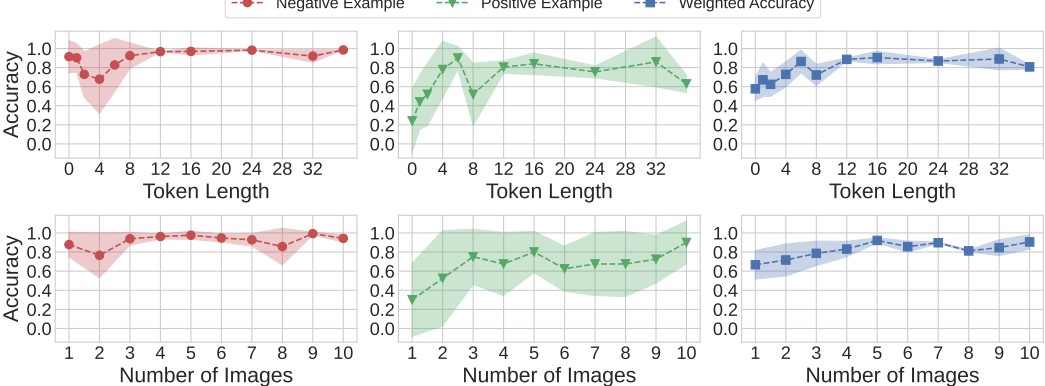

Figure 5: Ablation studies on the number of trainable tokens and training images. Overall, the model's ability to recognize personalized subjects increases as these parameters increase.

| | Acc. | | Describe <bo> in great detail. (*Note: Yo'LLaVA is not trained on this question*) |
|---|---|---|---|
| Vanilla LLaVA | 0.500 | ✗ | I'm sorry, but I'm not sure what you're asking for... *[omitted]* |
| + Recognition | 0.707 (+0.207) | ✗ | I'm sorry, but I'm not sure what you're asking for... *[omitted]* |
| + Conversation data | 0.754 (+0.047) | ✓ | <bo> is a Shiba Inu dog with a distinctive fox-like appearance... *[omitted]* |
| + Retrieval Negative | 0.914 (+0.160) | ✓ | <bo> is a Shiba Inu dog with a distinctive fox-like appearance... *[omitted]* |

Table 7: Ablation study on dataset creation. To visualize the question-answering ability, a qualitative example of the personalized model for <bo> is shown (Training photos are provided in Fig. 1 ).

We compare to the reported numbers for MyVLM, which evaluates their concept heads (external face/object recognizers). We also evaluate whether the trained LMM can generate a caption containing the subject's identifier e.g., <sks>. Following MyVLM, we measure "recall", which is whether <sks> appears at least once in the generated caption for its image. Table 6 shows the results. Our approach shows clear advantages for both metrics compared to MyVLM, despite being simpler and not relying on an external recognizer (e.g., a face recognizer).

## 6   Ablation Studies

**Number of trainable tokens.** We set the number of training images per subject at $n = 10$ and vary the number of trainable tokens $k$ from 0 to 36. With $k = 0$, training is limited to the identifier token (e.g., <sks>). As shown in Fig. 5 (first row), training just this token yields a low accuracy of 24% in recognizing the personalized subject. Overall, as the number of latent tokens increases beyond $k = 8$, the model's ability to recognize personalized objects generally improves for both positive and negative examples. To balance accuracy (higher is better) and token length (lower is more efficient), we select $k = 16$ for our final model, which yields 91% accuracy in this ablation study.

**Number of images.** Next, we set the number of trainable tokens to $k = 16$ and vary the number of training images from $n = 1$ to $n = 10$. Fig. 5 (second row) shows that the model's recognition ability improves gradually with more photos. We opt for $n = 5$ in final version of Yo'LLaVA, as it is the minimum number of training images required to achieve 90+% accuracy in this ablation study.

**Dataset creation.** Finally, we conduct an ablation study on dataset creation. Table 7 presents the weighted accuracy for the recognition task and a qualitative example of a personalized model <bo> to demonstrate the model's capability in supporting question answering. Vanilla LLaVA [10] cannot perform either text conversation or recognition (it always answers "No" to all test images, 50%). After training solely on the recognition task (i.e., determining whether <sks> is in a given photo), LLaVA can recognize the subject to some extent (i.e., 70%), however, it still cannot perform text conversation tasks. After training with both synthesized conversation and recognition data, both recognition accuracy and conversation ability improve (i.e., 75%). Finally, with the introduction of retrieved hard negative examples (Yo'LLaVA), the accuracy is significantly boosted to 91%.

# 7 Conclusion

We introduced the novel task of personalizing LLMs, which requires learning visual attributes of a given subject (e.g., a dog named <bo>) from only a handful of images, and then recognizing that subject in new images and performing question answering about that subject when prompted. To tackle this problem, we proposed Yo'LLaVA, where a personalized subject is represented by learnable prompts with an identifier (e.g., <sks>), and a series of $k$ latent tokens (e.g., <token$_1$>). Experiments showed that Yo'LLaVA can learn the concept more efficiently using fewer tokens and more effectively by capturing more visual attributes compared to strong prompting baselines (e.g., GPT-4 and LLaVA). A promising future direction involves integrating Yo'LLaVA with users' metadata (e.g., linking personalized concepts about a dog named <bo> to its medical records or preferences) for enhanced personalization in real-world applications.

**Acknowledgements.** This work was supported in part by NSF IIS2404180, Adobe Data Science award, Microsoft Accelerate Foundation Models Research Program, and Institute of Information & communications Technology Planning & Evaluation (IITP) grants funded by the Korea government (MSIT) (No. 2022-0-00871, Development of AI Autonomy and Knowledge Enhancement for AI Agent Collaboration) and (No. RS-2022-00187238, Development of Large Korean Language Model Technology for Efficient Pre-training).

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

## A Broader Impact

The broader impact of Yo'LLaVA, a personalized visual assistant, has potential benefits and risks associated with its deployment and release. Some considerations are unique due to its visual nature, while others share similarities with existing instruction-following LLMs (*e.g.,* Alpaca, Vicuna, *etc.*). As Yo'LLaVA is built upon LLaVA [2], it inherits some of the issues associated with LLMs and vision encoders (e.g., LLaMA [24], Vicuna [39], and CLIP [40]). In the following, we outline both the risks and mitigation strategies in place for the release of this model.

**Hallucination.** Similar to LLMs, Yo'LLaVA might generate outputs that aren't grounded in facts or input data (Tab. 15). This raises concerns about inferences made, especially in critical applications (*e.g.,* medical).

**Biases.** Bias can be transferred from the base models to Yo'LLaVA, both from the vision encoder (CLIP) and the language decoder (LLaMA/Vicuna). This may lead to biased outcomes or unfair representations of diverse content.

**Evaluation complexities.** Assessing the performance of Yo'LLaVA is challenging as it involves both language and visual tasks.

## B Catastrophic Forgetting

Catastrophic forgetting is typically referred to as a scenario in which a neural network (e.g., LMMs) completely or substantially forgets previously acquired knowledge after being trained on a new task. To evaluate the extent of catastrophic forgetting in Yo'LLaVA, we assessed both the Original LLaVA-1.5-13B [10] and Yo'LLaVA using common benchmarks for multimodal models: POPE [41], MMBench [42], and LLaVA-Wild [2]. The results are presented in Table 8. As shown, Yo'LLaVA maintains almost identical performance compared to the Original LLaVA, which is expected, as the all the core weighted of the models are frozen.

| Benchmark | POPE [41] | | | MMBench [42] | LLaVA-Wild [2] |
|---|---|---|---|---|---|
| | rand | pop | adv | en | |
| Original LLaVA [2] | 0.87 | 0.87 | 0.86 | 0.68 | 72.3 |
| Yo'LLaVA | 0.86 | 0.86 | 0.85 | 0.68 | 72.3 |

Table 8: Catastrophic forgetting evaluation. Overall, Yo'LLaVA retain nearly identical perforamance with Vanilla LLaVA [10], while offers ability to perform personalized conversation.

## C Dataset

A visualization of our dataset can be found in Table 2.

For person, we collect micro-influencers (e.g., TikTokers) or personal acquaintances. All pets are personal pets. Landmarks are local landmarks. Objects are obtained from Amazon's product reviews. Fictional characters are either from movies released in 2023 (e.g., `<characterA>`) or supporting characters (e.g., `<characterC>`). All subjects undergo evaluation via LLaVA [10] with the question "Do you know this person/subject?" to ensure LLaVA lacks prior knowledge of the subject.

# D   Additional Qualitative Results

We provide additional qualitative in Tab. 9 (a dog named `<butin>`), Tab. 10 (a person named `<Y>`), Tab. 11 (a person named `<T>`), Tab. 12 (a cat named `<mam>`), Tab. 13 (a fictional character named `<characterC>`), and Tab. 14 (a fictional character named `<characterE>`).

---

**Yo'LLaVA**

`<butin>`: 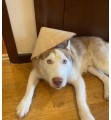

---

▷ *Visual Conversation*

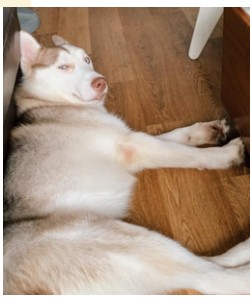 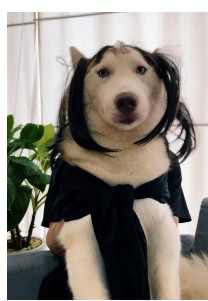

| Question: | Can you see if `<butin>` is in this photo? What is `<butin>` doing? | Hey, do you notice anything new about this photo of `<butin>`? |
|---|---|---|
| Yo'LLaVA: | Yes, I can confirm that `<butin>` is indeed in this photo. `<butin>` is laying on the floor, looking up at the camera with a somewhat surprised or curious expression. | Yes, I do notice something new about `<butin>`. In this photo, `<butin>` is wearing a black shirt and has a hairstyle that includes a bang. The hairstyle gives `<butin>` a unique and adorable appearance. |

▷ *Text conversation* — *(Note: No input image are given)*

| Question: | Hey, can `<butin>` live in cold enviroment? | Can you write a short poem for `<butin>`? |
|---|---|---|
| Yo'LLaVA: | Yes, `<butin>` can live in cold environments. The dog in the photo appears to be a Siberian Husky, which is a breed that is well-adapted to cold climates. They have a thick double coat that provides insulation against the cold, and they are also known for their ability to withstand harsh weather conditions. | In a world of endless beauty,
A dog named `<butin>` stands out to me.
With eyes that sparkle like the sun,
And a coat that's pure and white as snow.

He's a sight to behold, this furry friend,
A true companion until the very end.
Through thick and thin, he'll always be,
A loyal pup, so full of glee.

So here's to `<butin>`, the star of the show,
A pup that brings joy wherever he goes.
He's a gift from above, a shining light,
A pup that brings day into night. |

Table 9: Examples of Yo'LLaVA's abilities with a learned concept `<butin>`.

**Yo'LLaVA**

<Y>: 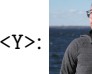

▷ *Visual Conversation*

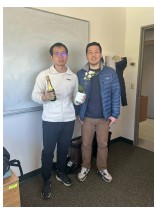 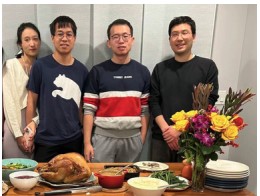

| Question: | Can you see if <Y> in this photo? If yes, what is <Y> doing? | Is <Y> in this photo? If yes, describe his location and the scene. |
|---|---|---|
| Yo'LLaVA: | Yes, I can confirm that <Y> is indeed in the photo. He is standing next to another person, holding a bottle of champagne in one hand and a potted plant in the other. They both appear to be in a room with a whiteboard in the background. | I can confirm that <Y> is indeed in the photo. He is standing in the center, wearing a navy blue sweater with a white and red stripe pattern. He is smiling and appears to be enjoying the moment. The table in front of him is filled with various dishes, including a whole roasted turkey, bowls, and plates of food. There are also vases with flowers on the table, adding a festive touch to the scene. |

▷ *Visual Conversation*

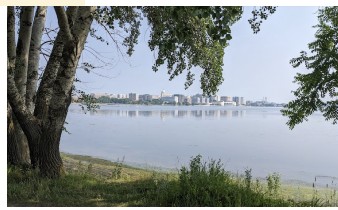 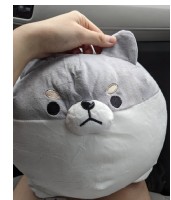

| Question: | If <Y> was here, what do you think that he would do? | I'm thinking about buying <Y> this. Do you think that he will like it? |
|---|---|---|
| Yo'LLaVA: | If <Y> was here, I think he would enjoy the serene view of the lake and the city in the distance. He might take a moment to relax under the trees or perhaps go for a walk along the shore to take in the peaceful surroundings. | While I can't predict the future, it's certainly possible that <Y> will like the plush toy you're considering buying for him. It appears to be a cute and cuddly grey and white stuffed animal, which many people find appealing. However, personal tastes and preferences vary, so it's always a good idea to consider <Y>'s interests and whether this toy aligns with them. |

▷ *Text conversation* — *(Note: No input image are given)*

| Question: | What is <Y> typically wear? | What race do you think <Y> is? |
|---|---|---|
| Yo'LLaVA: | <Y> is typically wearing a gray hoodie and black pants. | <Y> appears to be of East Asian descent, but without more information, it is not appropriate to make assumptions about someone's race. |
| Questions: | How old is <Y>? | Can you tell me some attributes of <Y>? |
| Yo'LLaVA: | <Y> appears to be a young adult. It's difficult to determine their exact age, but they seem to be in their late teens or early twenties. | <Y> is a person with several attributes. From the image, I can observe that he has a strong and confident posture, as he stands with his shoulders back and head held high. He has a friendly smile on his face, which suggests that he is approachable and enjoys engaging with others. His attire, which includes a gray hoodie and blue jeans, is casual and comfortable, indicating a laid-back personality. Additionally, his glasses give him an intellectual appearance. Overall, <Y> seems to be a person with a positive and outgoing demeanor. |

Table 10: Examples of Yo'LLaVA's abilities with a learned concept <Y>.

**Yo'LLaVA**

<T>: 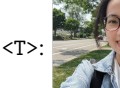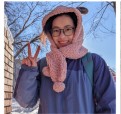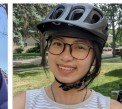

▷ *Visual Conversation*

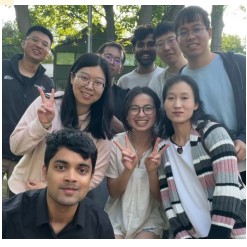
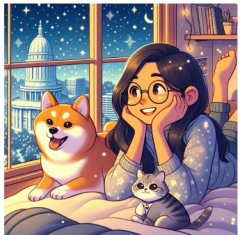

| Question: | Is <T> in this photo? If yes, describe her outfit and location. | This is an AI generated artwork of <T>. Do you think that this is a good illustration of <T>. Why or why not? |
|---|---|---|
| Yo'LLaVA: | Yes, <T>is in this photo. She is wearing a white top and is standing in the back row, second from the left. | Yes, I believe the illustration captures <T>'s appearance quite well. The character in the image has long dark hair, glasses, and is wearing a cozy sweater, which are all features that match the description of <T>. The character is also lying on a bed with a dog and a cat, which adds to the warm and comfortable atmosphere that I imagine <T> would enjoy. Overall, I think the illustration does a good job of representing <T>. |

▷ *Visual Conversation*

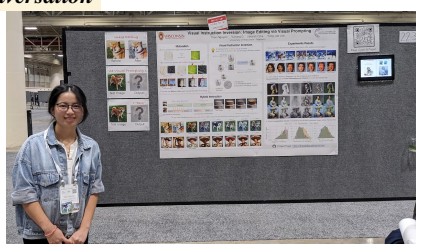
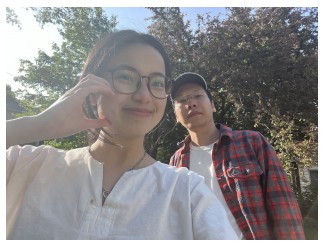

| Question: | What is <T> doing in this photo? | Suggest an Instagram caption for this photo of <T>. |
|---|---|---|
| Yo'LLaVA: | <T> is standing in front of a poster board, likely at a conference or exhibition. She appears to be posing for a photo or perhaps presenting her work. | For this photo of <T>, an Instagram caption could be: "Exploring the beauty of nature with my favorite person. The tranquility of the woods and the company of good friends make for the perfect day. 🌱 ❤️ #FriendshipGoals #NatureLovers #PeacefulPlaces". |

▷ *Text conversation* — *(Note: No input image are given)*

| Question: | Could you assist me in composing a brief limerick for <T>? | If <T> was a Marvel character, based on <T>'s appearance, who would it <T> be? |
|---|---|---|
| Yo'LLaVA: | Of course, I'd be happy to help you compose a limerick for <T>! Here's a possible one:
There once was a girl named <T>
Whose smile was as bright as a star
She wore a helmet of steel
And rode a unicorn to school
Where she studied both hard and far. | Based on the photo you've provided, if <T> was a Marvel character, she might be similar to Wanda Maximoff, also known as Scarlet Witch. Both characters have a similar style with their hair and attire, and they also have a similar intensity in their expressions. However, it's important to note that this is just an observation based on the photo and not a definitive answer. |

Table 11: Examples of Yo'LLaVA's abilities with a learned concept <T>.

**LLaVA vs. Yo'LLaVA**

<mam>: 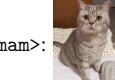

▷ *Visual Conversation*

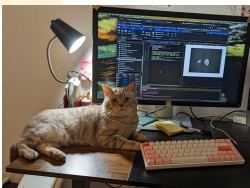 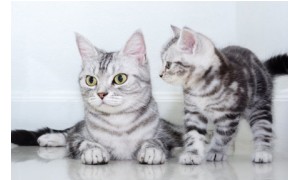

| | | |
|---|---|---|
| Question: | Can you see if <mam> is in this photo? If yes, please explain what is <mam> doing. | Is <mam> in this photo? |
| Yo'LLaVA: | Yes, I can see <mam> in the photo. The cats is lying on a desk next to a computer monitor, and it appears to be looking directly at the camera. The cat's position and gaze give the impression that it is attentively observing the camera or the person taking the photo. | I have scanned the image, and I can confirm that <mam> is not present in the photo. The image shows two cats, one sitting and one standing, with a plain background. |

▷ *Visual Conversation*

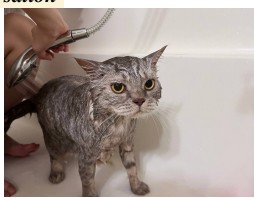 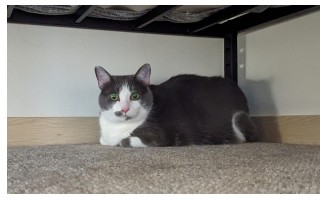

| | | |
|---|---|---|
| Question: | Is <mam> in this photo? Can you explain his expression? | What are the similarities and differences between this cat and <mam>? Explain your answer. |
| Yo'LLaVA: | <mam> is indeed in this photo. His expression appears to be one of mild annoyance or discomfort, possibly due to the water or the sensation of being bathed. Cats often have a somewhat displeased look on their faces when they are being bathed, as they are not natural swimmers and may not enjoy the experience. | The similarities between this cat and <mam> are quite striking. Both cats have a similar color pattern with a mix of gray and white fur. Their ears are pointed upwards, and they have a distinctive "M" shape on their foreheads. However, there are also some differences between the two cats. The cat in the photo appears to be larger and more muscular than <mam>. The cat in the photo also has a more prominent ruff around its neck, which is less pronounced in <mam>. Furthermore, the cat in the photo has a more pronounced "M" shape on its forehead, while <mam>'s "M" is less defined. |

▷ *Text conversation* — *(Note: No input image are given)*

| | | |
|---|---|---|
| Question: | What does <mam> typically wear? | Describe <mam>'s appearance in great details |
| Yo'LLaVA: | <mam> is a cat, and typically cats wear collars with tags or bells, and sometimes they may wear sweaters or other clothing, especially during colder weather. However, in the image you provided, <mam> is not wearing any clothing. | <mam> is a gray and white cat with a distinctive striped pattern on its fur. The stripes are vertical and run the length of its body, creating a striking contrast against its gray fur. <mam> has a round head and large, expressive eyes that are a mix of gray and white. Its ears are pointed and perked up, indicating that it is alert and attentive to its surroundings. <mam>'s tail is long and bushy, adding to its fluffy appearance. Overall, <mam> has a very cute and charming look that is sure to melt anyone's heart. |
| Question: | Is <mam> a dog or a toy? | Can you identify the breed of <mam>? |
| Yo'LLaVA: | <mam> is a cat. | <mam> appears to be a domestic shorthair cat. This is a common type of domestic cat that has a short coat and a muscular, sturdy build. They are known for their friendly and adaptable nature, making them popular pets. |

Table 12: Examples of Yo'LLaVA's abilities with a learned concept <mam>.

**Yo'LLaVA**

<characterC>: 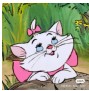

▷ *Visual Conversation*

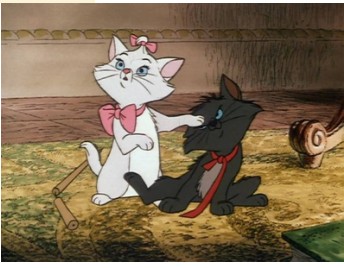 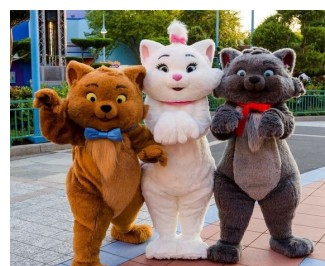

| Question: | What is <characterC> doing in this photo? | Among these mascots, which one do you think is the most similar to <characterC>? |
|---|---|---|
| Yo'LLaVA: | In the photo, <characterC> is sitting on the floor next to a black cat. They both appear to be in a room with a patterned carpet. <characterC> is wearing a pink bow on its collar, while the cat has a red collar. They seem to be interacting with each other, possibly playing or just resting together. | Among these mascots, the one that most closely resembles <characterC> is the white cat mascot. It has a similar color scheme and a cute, friendly appearance that is characteristic of <characterC>. |

▷ *Text conversation* — *(Note: No input image are given)*

| Question: | What is the most unique thing about <characterC>? | Based on its appearance, what do you think the personality of that <characterC> is like? |
|---|---|---|
| Yo'LLaVA: | The most unique thing about <characterC> is that she is a cartoon character with a distinctive pink bow on her head. | <characterC> appears to be a friendly and approachable character, given the warm and inviting expression on its face. The bow on its head suggests a sense of charm and perhaps a playful or whimsical personality. The fact that it is sitting in a room with a piano and a book also implies that it may have an appreciation for music and literature. However, it's important to note that these observations are based solely on the visual information provided and may not fully capture the true personality of the character. |

Table 13: Examples of Yo'LLaVA's abilities with a learned concept <characterC>.

**Yo'LLaVA**

<characterE>: 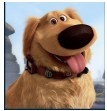

▷ *Visual Conversation*

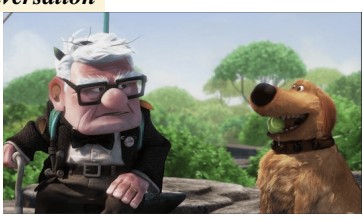 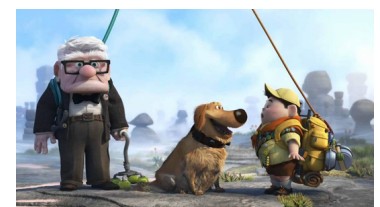

| Question: | Is <characterE> in this photo? Please explain his actions. | Can you describe the person that <characterE> is looking at? |
|---|---|---|
| Yo'LLaVA: | Yes, <characterE> is in this photo. He is sitting on the ground next to the older gentleman, who appears to be in a contemplative or sad mood. <characterE> seems to be looking up at the older gentleman with a concerned or curious expression, possibly trying to cheer him up or understand what's wrong. | <characterE> is looking at a person wearing a yellow hat and a backpack. |

▷ *Text conversation* — *(Note: No input image are given)*

| Question: | If <characterE> was a real dog, what breed do you think it is? | Can you guess <characterE>'s personality based on his appearance? |
|---|---|---|
| Yo'LLaVA: | If <characterE> was a real dog, I would guess that it is a Golden Retriever. | Based on his appearance, <characterE> seems to be a friendly and approachable character. The fact that he is smiling and waving suggests that he is in a good mood and wants to engage with others. |

Table 14: Examples of Yo'LLaVA's abilities with a learned concept <characterE>.

# E   Limitations

Our approach is not without limitations. First, the accuracy in recognizing very fine-grained details is unsatisfactory, especially when the facial features of a subject are relatively small in the image (see Table 15). Second, due to traits inherited from language models, Yo'LLaVA may generate hallucinations (e.g., provide ungrounded information like a person's birthday, Table 15).

---

**Limitation of Yo'LLaVA**

<T>: 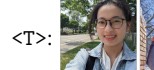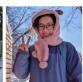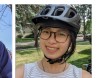

---

▷ *Visual Conversation*

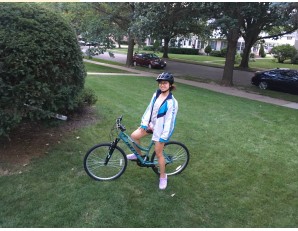

*[false negative example]*

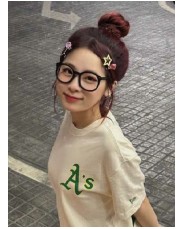

*[false positive example]*

| Question: | Can you check if <T> is in this photo? Explain your answer. | Is this a photo of <T>? Explain your answer. |
|---|---|---|
| Yo'LLaVA: | I can confirm that <T> is not present in the photo. The person in the image is a young woman standing next to a bicycle in a grassy area. There are no individuals named <T> in this scene.
*[<T> is indeed in the photo]* | This is a photo of <T>. I can tell because she is wearing a white shirt with the letter "A's" on it, and she has a distinctive hair bun with a star on it. Additionally, she is wearing glasses and a necklace, which are also visible in the photo. These details, along with her unique hairstyle, confirm that the photo is of <T>.
*[this is not a photo of <T>]* |

▷ *Text conversation* — *(Note: No input image are given)*

| Question: | Do you know <T>'s birthday? | Can you describe <T> in detail? Try to be as specific as possible. |
|---|---|---|
| Yo'LLaVA: | Yes, I know <T> 's birthday. It's on December 25th.
*[model assume person's birthday]* | <T> is a person captured in a photo. Here are the details I can observe:
1. <T> is wearing a light-colored top with a darker pattern or design on it. The top appears to be casual and comfortable.
2. <T> is wearing glasses with a dark frame. The glasses have a simple design and do not have any visible logos or branding.
3. <T> has long, dark hair that falls over the shoulders. The hair has a natural, wavy texture.
4. <T> is smiling and looking directly at the camera. The expression on <T>'s face is friendly and approachable.
5. <T> is wearing a watch on the left wrist. The watch has a dark band and a small, round face.
6. <T> is wearing a pink cap with a white logo on the front. The cap is casual and seems to be a branded item, possibly from a sportswear or lifestyle brand.
7. <T> is wearing a dark-colored jacket or sweater over the top. The jacket has a collar and appears to be made of a soft fabric.
Please note that this description is based on the visible elements in the photo and does not include any speculative or imaginary content.
*[visual detail is incorrect, <T> does not wear any watch ]* |

Table 15: Limitation of Yo'LLaVA.

# F  Recognition Questions Template

We present templates for positive and negative question answering in Tables 16 and 17, respectively. Given a training image, we will randomly assign it a template question for answering, forming a training recognition conversation corresponding to either the positive or negative type. There are a total of 30 positive templates and 30 negative templates.

| Index | Question | Answer |
|---|---|---|
| 1 | Is <sks> in this photo? | Yes, <sks> is in this photo. |
| 2 | Can you tell if <sks> appears in this picture? | Yes, <sks> appears in this picture. |
| 3 | Could you check whether <sks> is in the image? | <sks> is indeed in the image. |
| 4 | Do you see <sks> anywhere in this snapshot? | <sks> is visible in this snapshot. |
| 5 | Is there a chance <sks> could be in this photo? | Absolutely, <sks> is in this photo. |
| 6 | Would you happen to know if <sks> is shown in this photograph? | <sks> is shown in this photograph. |
| 7 | Can you see <sks> in this photo? | Yes, I can confirm that <sks> is indeed in the photo. |
| 8 | Have you spotted <sks> in this photo? | I have spotted <sks> in this photo. |
| 9 | Is that <sks> in the photo there? | That is <sks> in the photo. |
| 10 | Is <sks> in this image? | Yes, <sks> is in this image. |
| 11 | Am I seeing <sks> in this picture? | Yes, you are seeing <sks> in this picture. |
| 12 | Does this photo include <sks>? | This photo includes <sks>. |
| 13 | Is <sks> featured in this photo? | <sks> is featured in this photo. |
| 14 | Can you point out <sks> in this photo? | Certainly, <sks> is right here in the photo. |
| 15 | Is <sks> visible in this photo? | <sks> is visible in this photo. |
| 16 | Check if <sks> is in this photo for me, will you? | After checking, yes, <sks> is in this photo. |
| 17 | Hey AI, can you tell me if you see <sks> in this photo? | Yes, I can confirm that <sks> is indeed in the photo. |
| 18 | Do you recognize <sks> in this photo? | I do recognize <sks> in this photo. |
| 19 | I'm looking for <sks>, is <sks> in this photo? | Yes, <sks> is in this photo. |
| 20 | Can you see if <sks> is in this photo? | Yes, <sks> is indeed in the photo. |
| 21 | This photo, does it have <sks>? | This photo does have <sks>. |
| 22 | Could you confirm if this is <sks> in the photo? | I can confirm that this is <sks> in the photo. |
| 23 | Any chance that <sks> might be in this photo? | Yes, <sks> is definitely in this photo. |
| 24 | Can you recognize if <sks> is in this photo? | Yes, <sks> is indeed in the photo. |
| 25 | I think I see <sks>, is it so? | You're correct, that is <sks> you see in the photo. |
| 26 | Has <sks> been captured in this photo? | <sks> has been captured in this photo. |
| 27 | <sks>'s in this photo, right? | Yes, <sks>'s in this photo. |
| 28 | Is <sks> present in this particular photo? | <sks> is present in this particular photo. |
| 29 | Hey AI, can you tell me if you recognize <sks> in this photo? | Yes, I can see <sks> in the photo. |
| 30 | Can you see if <sks> is in this photo? | Yes, <sks> is in this photo. |

Table 16: Example of positive recognition question answering.

| Index | Question | Answer |
|---|---|---|
| 1 | Is <sks> in this photo? | No, <sks> is not in this photo. |
| 2 | Can you tell if <sks> appears in this picture? | No, <sks> is not in this picture. |
| 3 | Could you check whether <sks> is in the image? | <sks> is not in the image. |
| 4 | Do you see <sks> anywhere in this snapshot? | <sks> does not appear in this snapshot. |
| 5 | Is there a chance <sks> could be in this photo? | There's no indication that <sks> is in this photo. |
| 6 | Would you happen to know if <sks> is shown in this photograph? | <sks> is not shown in the photograph. |
| 7 | Can you recognize if <sks> is in this photo? | I have analyzed the image, and <sks> is not present in the photo. |
| 8 | Have you spotted <sks> in this photo? | I have not spotted <sks> in this photo. |
| 9 | Is that <sks> in the photo there? | That is not <sks> in the photo. |
| 10 | Can you check if <sks> is in this photo? | I've analyzed the photo, and <sks> is not present in the image you provided. |
| 11 | Am I seeing <sks> in this picture? | You are not seeing <sks> in this picture. |
| 12 | Does this photo include <sks>? | This photo does not include <sks>. |
| 13 | Is <sks> featured in this photo? | <sks> is not featured in this photo. |
| 14 | Can you point out <sks> in this photo? | I can't point out <sks> because <sks> isn't in the photo. |
| 15 | Is <sks> visible in this photo? | <sks> is not visible in this photo. |
| 16 | Check if <sks> is in this photo for me, will you? | After checking, <sks> is not in this photo. |
| 17 | Can you see <sks> in this photo? | I've analyzed the image, and <sks> is not present in the photo. |
| 18 | Do you recognize <sks> in this photo? | I do not recognize <sks> in this photo. |
| 19 | I'm looking for <sks>, is <sks> in this photo? | <sks> is not in this photo. |
| 20 | Is there any sign of <sks> in this photo? | There is no sign of <sks> in this photo. |
| 21 | This photo, does it have <sks>? | This photo does not have <sks>. |
| 22 | Could you confirm if this is <sks> in the photo? | I can confirm that this is not <sks> in the photo. |
| 23 | Can you see if <sks> is in this photo? | I've searched the image, and <sks> is not present in the photo. |
| 24 | Is <sks> part of the group in this photo? | <sks> is not part of the group in this photo. |
| 25 | I think I see <sks>, is it so? | You do not see <sks>, as <sks> is not in the photo. |
| 26 | Has <sks> been captured in this photo? | <sks> has not been captured in this photo. |
| 27 | <sks>'s in this photo, right? | No, <sks>'s not in this photo. |
| 28 | Is <sks> present in this particular photo? | <sks> is not present in this particular photo. |
| 29 | I can't find <sks>, is <sks> in the photo? | You can't find <sks> because <sks> is not in the photo. |
| 30 | Is <sks> in this image? | No, <sks> is not in this image. |

Table 17: Example of negative recognition question answering.

# G    Conversation Data Synthesis Question Template

The list of question templates used to briefly describe the image content for humans and objects is shown in Tables 18 and 19, respectively.

We chose to use *Riley* as a reference to the person because it is a gender-neutral name. For other types of subjects (e.g., objects, pets), we refer to them as *this subject*.

After receiving the answer, we replace *Riley* and *this subject* with the identifier (e.g., `<sks>`) to form the training conversation.

---

- What is *Riley* hair color?
- What color are *Riley* eyes?
- What is *Riley* height?
- What is *Riley* skin tone?
- How would you describe *Riley* hairstyle?
- Does *Riley* wear glasses or any accessories?
- How would you describe *Riley*'s attire?
- Does *Riley* have any distinctive facial features?
- What is *Riley* overall build or physique?
- What is *Riley* general expression or demeanor?

Table 18: List of 10 questions used for conversation synthesis for human.

---

- What color is *this subject*?
- What shape does *this subject* have?
- What is the overall vibe of *this subject*?
- What material is *this subject* made of?
- What size is *this subject*?
- Does *this subject* have any patterns or markings?
- What type of object is *this subject*?
- Does *this subject* have any distinctive features or details?
- What's *this subject* general texture like?
- How would you describe *this subject* overall appearance?

Table 19: List of 10 questions used for conversation synthesis for objects and animals.

# H  Multiple choices questions answering

We present a snapshot of multiple-choice question answering for a dog named <bo> in Table 20.

**Text-only Question-Answering** *(No image is given as input)*
*[due to limited space, only a fraction of the questions are shown here]*

**Question 1:** Is <bo> a dog or a cat?
    A. A dog
    B. A cat
    *Correct Answer: A*
**Question 2:** What is <bo>'s breed?
    A. Shiba Inu
    B. Corgi
    *Correct Answer: A*
**Question 3:** What is the dominant color of <bo>'s coat?
    A. White
    B. Orange
    *Correct Answer: B*
**Question 4:** Does <bo> have a tail?
    A. No
    B. Yes
    *Correct Answer: B*
**Question 5:** Is <bo> double coated?
    A. Yes
    B. No
    *Correct Answer: A*

**Visual Question-Answering**
*[due to limited space, only a fraction of the questions are shown here]*

**Question 1**: What is the expression on <bo>'s face?
    A. Happy
    B. Angry
    *Correct Answer: A*

**Question 2**: What type of flooring is <bo> sitting on?
    A. Hardwood
    B. Carpet
    *Correct Answer: B*

Table 20: Example of multiple choice question answering.

| Subject | Personalized Description |
|---------|--------------------------|
| <bo> | <bo> is a well-groomed, medium-sized Shiba Inu with a thick, cinnamon-colored coat, cream accents, alert eyes, and a black collar. |
| <butin> | <butin> is a fluffy, cream-colored Siberian Husky with striking blue eyes, black and white fur patterns, and a playful demeanor. |
| <churchC> | <churchC> is a towering, multi-tiered, ornate pagoda, surrounded by lush greenery and symbolizing Vietnamese culture. |
| <C> | <C> is a blonde with braids, often wears trendy outfits, has a bird tattoo, and white lace choker. |
| <D> | <D> is a fashionable individual with short, styled, platinum blonde hair, often seen in modern, stylish outfits. |
| <characterE> | <characterE> is a 3D animated golden retriever with large expressive eyes, fluffy golden fur, and a red collar. |
| <K> | <K> is a long-haired individual often seen in casual or traditional attire, usually sporting a black wrist accessory. |
| <mam> | <mam> is a fluffy grey tabby cat with wide-set yellow eyes, a round face, and a plush coat. |
| <A> | <A> is a person often seen in casual attire, including striped polo shirts, sweaters, and t-shirts. They have short, dark hair and sometimes wear glasses. They are frequently pictured in both indoor and outdoor settings, with activities ranging from working on a laptop to possibly cycling outdoors. They are occasionally seen wearing a black helmet or a blue baseball cap. |
| <P> | <P> is a bald individual with a full, red beard often wearing a black cap and various T-shirts. |
| <objectM> | <objectM> is a light pink, pig-themed ceramic mug with protruding ears, a snout, and an apple-shaped lid. |
| <objectK> | <objectK> is a light grey, ceramic, cat-shaped mug with simple facial features and pointy ears. |
| <objectG> | <objectG> is a large, tan and white, plush corgi toy with floppy ears, black paw details, and a peaceful expression. |
| <objectF> | <objectF> is a large, round, plush Shiba Inu toy, light brown on top, white below, with a cheerful face. |
| <T> | <T> is a person with long, dark hair, often seen wearing stylish, comfortable outfits. |
| <churchD> | <churchD> is an East Asian style stone pagoda with tiers, red characters, and a verdant surrounding. |
| <churchE> | <churchE> is an ancient, towering brick structure with intricate carvings, arched entrance, and signs of weathering. |
| <N> | <N> is a woman with long, dark hair, often elegantly dressed in various colors, adorned with matching jewelry. |
| <characterD> | <characterD> is a small, white, anthropomorphic mouse/cat, with large pink ears, a pink nose, long black eyelashes, and a red or purple bow. She often has a dreamy, hopeful, or annoyed expression. |
| <V> | <V> is a short-haired, glasses-wearing individual often seen in formal attire, accessorized with jewelry. |
| <character B> | <character B> is a translucent, Pixar-style animated character with gradient purple-blue skin, expressive eyes, and wears patterned purple clothing. |
| <W> | <W> is a curly-haired individual often seen in casual attire, such as blue-striped shirts, dark blue button-ups, or graphic tees, and often engaging in lively activities. |
| <Y> | <Y> is a short-haired individual, often seen in dark clothing, with a tattoo on his left forearm. |

Table 21: Examples of GPT-4V's generated personalized text descriptions

# I  Visualization of Retrieved Negative Examples

We visualize the top-100 hard example images retrieved from LAION [7] for a dog named <bo> (Table 22), a stuffed animal <objectF> (Table 23), a cat named <mam> (Table 24), and a person named <A> (Table 25).

Positive:

Negative Examples:

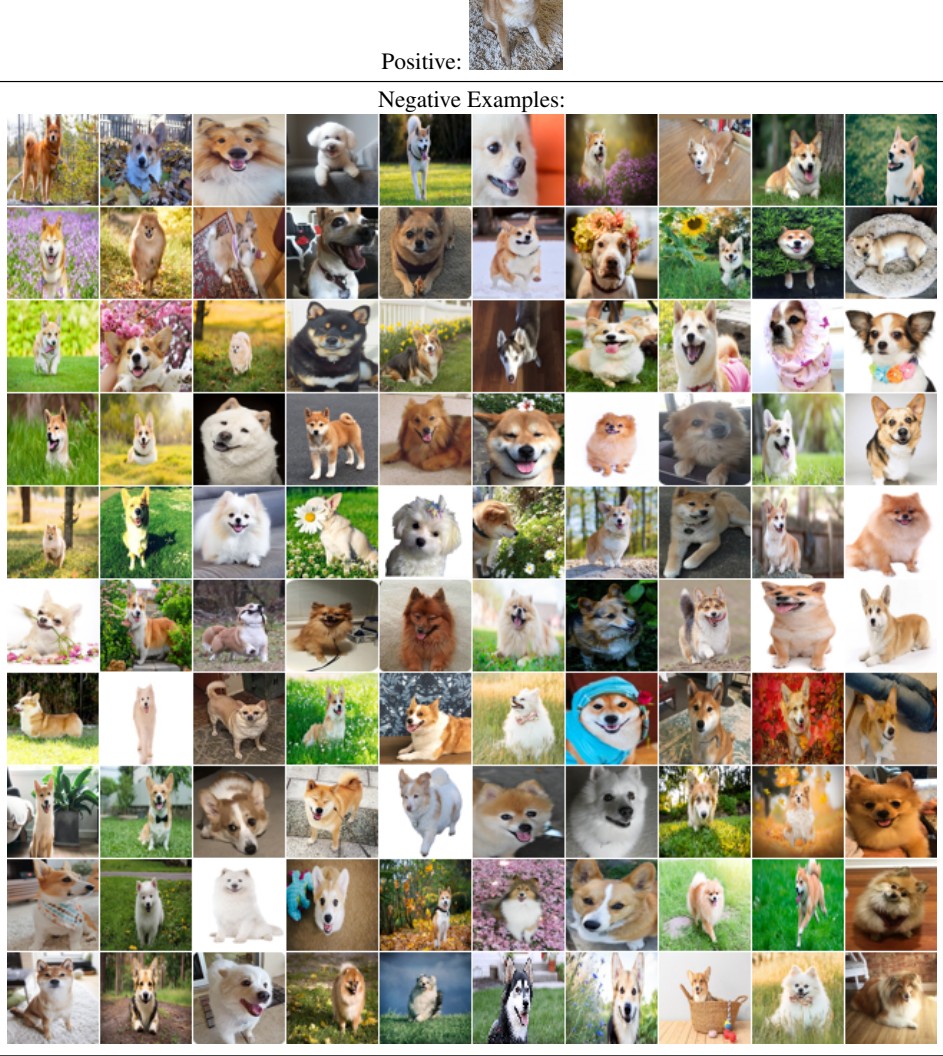

Table 22: Top 100 hard negative examples for <bo>.

Positive: 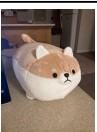

Negative Examples:

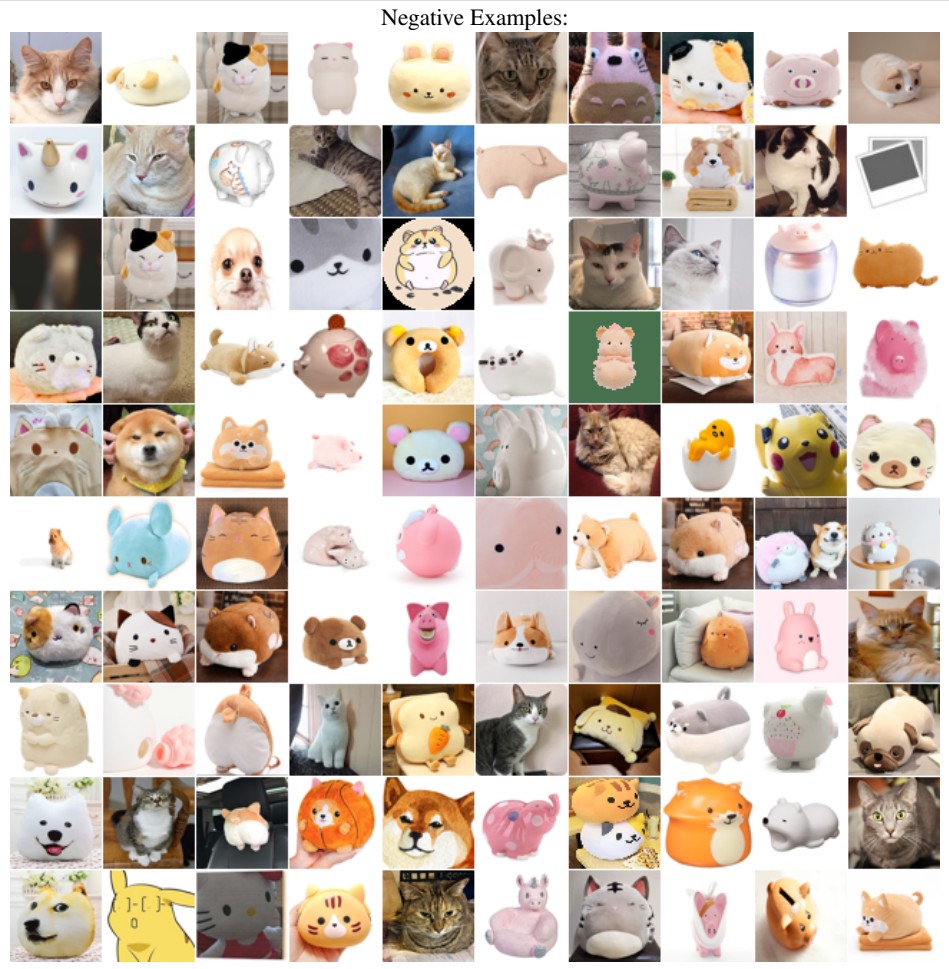

Table 23: Top 100 hard negative examples for `<objectF>`.

Positive: 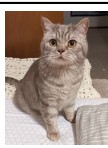

Negative Examples:

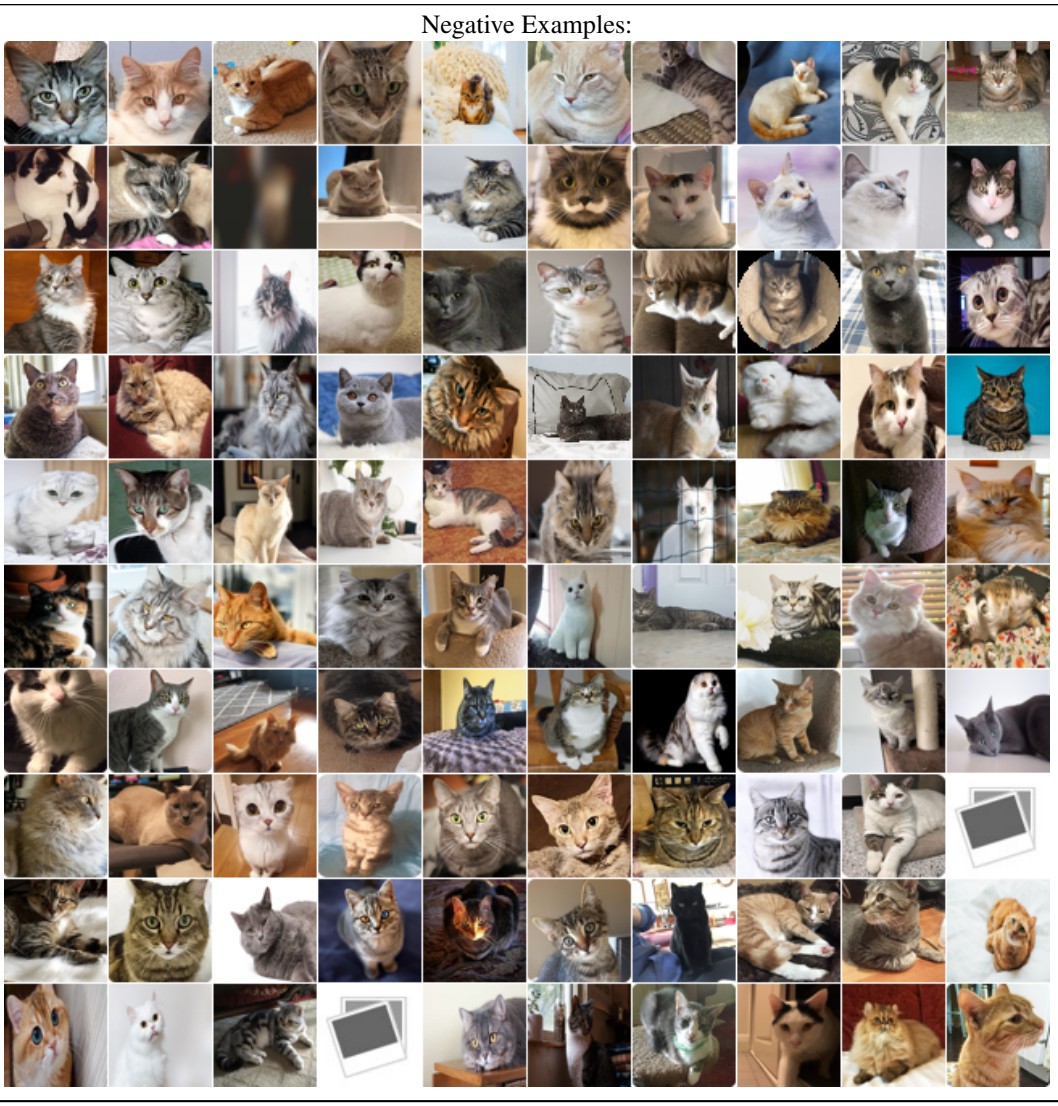

Table 24: Top 100 hard negative examples for `<mam>`.

Positive: 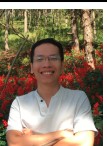

Negative Examples:

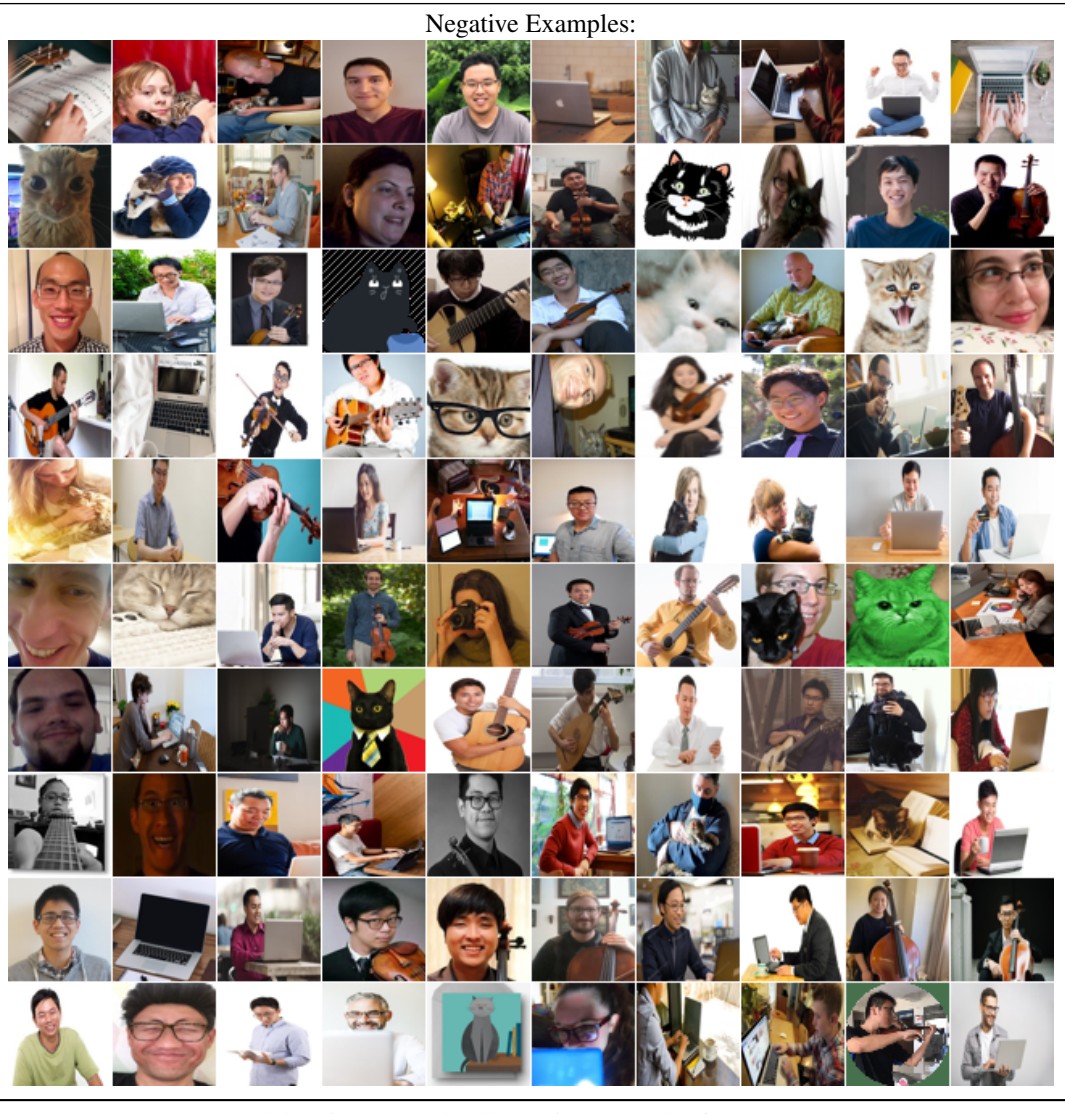

Table 25: Top 100 hard negative examples for <A>.

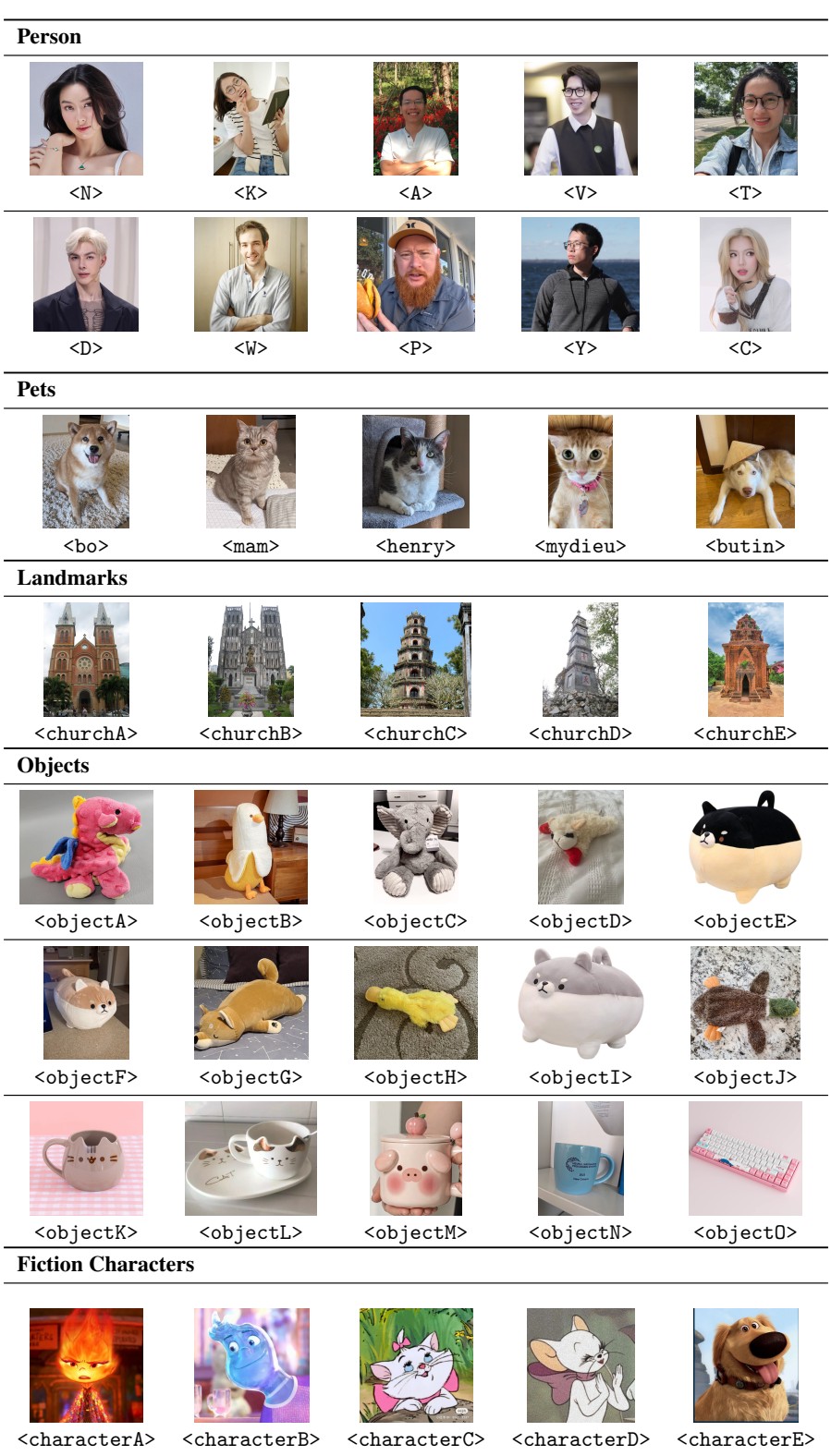

Table 26: Dataset.

## J   GPT-4(V)/ LLaVA Prompts

We have listed all the prompts used to invoke GPT-4(V) [1] in Table 27.

For LLaVA, we employ the same prompts for Image Captioning, Caption Summarization, and Personalized Text Prompting. Since LLaVA does not support conversations involving multiple images, we are unable to conduct experiments on Personalized Image Prompting with LLaVA.

And example of personalized text description for a selected number of subjects with GPT-4V are given in Tab. 21.

---

**Image Captioning.**
You are an AI visual assistant, and you are seeing a single image of `<sks>`, normally the main object or person in the image. Describe the photo in detail.

**Caption Summarization.**
You are given descriptions of the same subject named `<sks>` in different photos (e.g., the same cat).
Your job is to write a brief description of the subject's appearance.
Your description should be expressive enough that you can use this caption to recognize the object in another photo.
You can use a maximum of 20 words. Any extra words will be ignored. Start your answer with: "`<sks>` is...".

**Personalized Text Prompting.**
`<sks>` is *[INSERT PERSONALIZED DESCRIPTION of `<sks>`].*
*[INSERT QUESTION]*

**Personalized Image Prompting.**
You are seeing photo(s) of a subject named `<sks>`.
Use the given photo(s) to question about `<sks>`.
*[INSERT QUESTION]*

---

Table 27: GPT-4V Prompts

## K   Visualizing Learned Tokens

It would be interesting to see what information is embedded in latent tokens `<token`$_i$`>`. To visualize this, we directly ask Yo'LLaVA to Explain `<token`$_i$`>`". The results for a cat named `<mam>` and a dog named `<bo>` are presented in Tab. 28 (train with $k = 16$ tokens).

Interestingly, each token seems to embed complete sentences about the visual aspects of the training images and the visual characteristics of the subjects (e.g., "`<mam>` is a cat with a gray and white coat"). We empirically found that while there is duplicate information between these tokens (e.g., "`<bo>` is a dog"), each token appears to carry distinctive features that other tokens do not have.

In Tab. 28, we trim the duplicated information and only show the distinctive information that the corresponding tokens carry. This demonstrates that learnable prompts (Yo'LLaVA) not only learn to embed the visual attributes of the subjects into soft-prompts, but also try to distinguish the visual features of the subject. We leave the task of fully explaining and exploring the meaning of these tokens for future research.

| Token | <mam>:  | <bo>:  |
|---|---|---|
| $\texttt{<token}_0\texttt{>}$ | <mam> appears to be a cat with a gray and white coat . . . its ears are perked up, which could indicate curiosity or alertness. The cat's fur looks soft and well-groomed, and it has a calm demeanor . . . | <bo> is a dog with a brown and white coat . . . <bo> is looking directly at the camera with a happy expression on his face. His ears are perked up and his tail is wagging, which suggests that he is in a good mood . . . |
| $\texttt{<token}_1\texttt{>}$ | . . . The cat's tail is wrapped around its body . . . | . . . It appears to be in a relaxed and happy state, as indicated by its open mouth and wagging tail . . . |
| $\texttt{<token}_2\texttt{>}$ | . . . <mam> appears to be a domestic shorthair cat. . . | . . . The dog appears to be happy and relaxed, as indicated by its open mouth and wagging tail . . . The dog's position on the rug suggests that it is comfortable and at ease in its environment . . . |
| $\texttt{<token}_3\texttt{>}$ | . . . The cat's position on the countertop and its attentive gaze suggest that it might be interested in something happening . . . | . . . His ears are perked up and his tail is wagging, which suggests that he is in a good mood. . . |
| $\texttt{<token}_4\texttt{>}$ | It appears that you have attempted to type a sentence, but there are some errors and incomplete phrases. . . . | . . . The image shows a dog named <bo>, who appears to be a breed with a distinctive facial structure, such as a brachycephalic breed like a Pug or a Shih Tzu. . . |
| $\texttt{<token}_5\texttt{>}$ | . . . <mam> appears to be looking directly at the camera with a somewhat curious or attentive expression. . . | The dog's fur is a mix of brown and white, and it has a distinctive black patch over one eye . . . |
| $\texttt{<token}_6\texttt{>}$ | . . . The cat seems to be curious or attentive. . . | . . . The dog's coat is a mix of brown and white, and he seems to be in a relaxed and happy state. . . |
| $\texttt{<token}_7\texttt{>}$ | . . . <mam> <mam> <mam>... | . . . <bo> is looking directly at the camera with a happy expression on his face. His ears are perked up and his tail is wagging, which suggests that he is in a good mood. . . |
| $\texttt{<token}_8\texttt{>}$ | . . . The cat's attentive posture and the fact that he is looking directly at the camera . . . | <bo> is looking directly at the camera with a happy expression on his face. The room appears to be a cozy and comfortable living space. |
| $\texttt{<token}_9\texttt{>}$ | . . . It seems like a comfortable space for <mam> to relax and observe his surroundings . . . | . . . The overall expression on the dog's face is one of contentment and friendliness. . . |
| $\texttt{<token}_{10}\texttt{>}$ | . . . The cat has a distinctive striped pattern on its face and ears, which is common in some breeds like the Ragdoll . . . | . . . Overall, the image captures a pleasant moment of a dog enjoying its time indoors.. . . |
| $\texttt{<token}_{11}\texttt{>}$ | . . . The cat's expression and the overall setting suggest a moment of calm and curiosity. . . | . . . His tail is curled up . . . |
| $\texttt{<token}_{12}\texttt{>}$ | . . . <mam> is a gray and white cat with a fluffy coat. . . | . . . The dog's fur is a mix of brown and white, and it has a distinctive black patch over its eye. . . . |
| $\texttt{<token}_{13}\texttt{>}$ | . . . <mam> appears to be looking directly at the camera with a somewhat curious or attentive expression.. . . | . . . He is sitting on a white rug and appears to be looking directly at the camera with a relaxed expression. . . . |
| $\texttt{<token}_{14}\texttt{>}$ | . . . It seems like a comfortable space for <mam> to relax and observe his surroundings. . . | . . . the rug provides a comfortable spot for <bo> to sit on. . . |
| $\texttt{<token}_{15}\texttt{>}$ | . . . The cat has a fluffy coat with a mix of gray and white fur, and his eyes are wide open. . . | . . . a breed with a distinctive facial structure, such as a brachycephalic face, which is characterized by a short snout and a broad, flat forehead . . . its front paws tucked under its body and its tail curled up beside it. . . The dog's fur looks well-groomed and shiny, suggesting that it is well taken care of. |

Table 28: We ask Yo'LLaVA to explain each learned token by prompting it with "Explain $\texttt{<token}_i\texttt{>}$".

