# OpenReview forum: "Yo'LLaVA: Your Personalized Language and Vision Assistant"
_NeurIPS.cc/2024/Conference — NeurIPS 2024 poster_

### Official Review · Reviewer_wZcX · 2024-07-03

**Soundness:** 4
**Presentation:** 4
**Contribution:** 4
**Rating:** 7
**Confidence:** 5

**Summary:**

This paper studies personalization for lage multimodal models (LMMs). More specifically, how can a model understand that "my dog" refers to the dog of the user. The proposed model, Yo'LLaVA, learns latent tokens to encode personalized subjects using a handful of images of for each personalized concept.

Regarding the methodology, the assumption is to have access of a small number of images (O(10)) of a person or a subject without any text label. This setting is realistic. In order to learn personalized concepts, the authors use one special token for the personalized concept (e.g., <sks>) and then k latent tokenss are used to capture relevant visual details. The personalized concept token is a new token, and the latent tokens soft ones.

During training, all parameters are frozen except the k+1 new tokens and the final classifier head. To capture fine-grained details in the latent tokens, the authors propose to use hard negative minig to gather negative examples that are visually similar but not identical to the personalized concept. More specifically, they generate conversational training data triplets (image, question, answer) using a template-based approach (10 manually written conversations). Negative samples are retrieved from LAION with the top m images with the highest CLIP image embedding similarity. Overall, they are n (training images) x m (hard negative images) + 100 (easy images) generated samples.

In the experiments, <10 images and 16 latent tokens are used to learn the personalization of a subject/person. A new dataset of 40 subjects is collected and used. The authors show the effectiveness of their approach on LLaVA-1.5-13B. The tasks are recognition ability and question answering. The proposed method significantly outperforms the baseline. I really appreciate that there is a comparison with a concurrent work MyVLM; this is definitely a plus that highlights the effectiveness of the proposed method. Finally, the ablation studies support the modeling decision. To close the loop, the authors could conduct a small human evaluation.

Overall, this paper is very well written, novel, and results are strong.

**Strengths:**

+ Novelty
+ Strong performance
+ Comparison with concurrent work

**Weaknesses:**

- While I acknowledge that personalized datasets don't exist, it would be nice to have datasets in other domains.
- Lack of a small human evaluation.

**Questions:**

Could you elaborate more regarding the computational time needed for your method?

**Limitations:**

Yes

---

> ### Author Rebuttal · Authors · 2024-08-06
>
> > **While I acknowledge that personalized datasets don't exist, it would be nice to have datasets in other domains.**
>
> Thanks for your suggestion. Our current dataset includes humans, pets, and objects for personalization. We anticipate that future research will introduce more datasets in other domains!
>
> > **Could you elaborate more regarding the computational time needed for your method?**
>
> The total time for learning a new subject with LLaVA-1.5-13B would be around 40 minutes.
> For a new subject with 3-5 images, the time required to create the training data is roughly 3 minutes (1-2 minutes for conversation creation and 1 minute for hard negative retrieval). The optimization process, including training new tokens, takes approximately 34 minutes for 15 epochs (measured on an A6000 machine).
>
> > **Lack of small human evaluation.**
>
> Thank you for your suggestions! Due to the limited time available during the rebuttal phase, unfortunately, we cannot include a human evaluation, as it would involve privacy concerns and obtaining permissions (e.g., from a person named <A>). We will continue to investigate this and report if it becomes feasible.

---

> > ### Comment · Reviewer_wZcX · 2024-08-12
> >
> > Thank you for your answers. I keep my score unchanged.

---

### Official Review · Reviewer_ddkC · 2024-07-06

**Soundness:** 3
**Presentation:** 4
**Contribution:** 4
**Rating:** 7
**Confidence:** 4

**Summary:**

This paper proposes a new task of adapting a LLaVA model on personal images on specific instances, e.g., a specific pet dog, and answer visual questions about the instance. The authors proposed a finetuning pipeline to learn identity tokens, and retrain the original LLaVA ability while being able accept the new identity tokens. Training and evaluation are done on a small datasets the authors collected. Experiments show the proposed method outperform LLaVA and GPT4o baselines, as well as an concurrent work.

**Strengths:**

- The task of personalized VLM is important and interesting. This paper setup the task and (small scale) datasets, and provide a valid first attempt to this task, with good results.

- The method of representing object identity by a few visual tokens, and train them on recognition tasks makes full sense to me. The method description is clear, and the effectiveness is shown in both the qualitative example and ablation studies.

- Quantitative results are strong and sufficient. The authors ablated the necessary design choices, and compared to concurrent works with similar goal on their benchmark.

- It is nice that the authors also verified the original LLaVA abilities are retrained (Table 8).

**Weaknesses:**

- From reading the paper, it is unclear to me if we need to train a separate model for each instance, or we can train a single model on N instances together using N*16 learned identity tokens. If it is the later, does the model has a number of objects limit that it can learn together?

- One limitation might be we always need to finetune the model to adapt to new instances (rather than in-context learning). However the finetuning cost seems to be low enough (1 A6000 GPU). More discussion on the finetuning cost (e.g., wall clock time) in the rebuttal is appreciated.

**Questions:**

Overall this paper works on an important new problem with a valid method. The method and evaluation all makes full sense to me. I only have clarification or discussion questions. I believe a large number of audience would be interested in this topic.

**Limitations:**

Yes.

---

> ### Author Rebuttal · Authors · 2024-08-06
>
> > **From reading the paper, it is unclear to me if we need to train a separate model for each instance, or we can train a single model on N instances together using N*16 learned identity tokens. If it is the later, does the model has a number of objects limit that it can learn together?**
>
> As stated in Sec. 3.1 (Lines 152-157), we train (1) the newly added tokens (e.g., $<sks>$) and (2) the final classifier head matrix associated with these newly added tokens.
> During training, we train for each subject separately; then at test time, we can easily adapt the model to handle N subjects by concatenating their personalized prompts together.
>
> For example, in Table 2 (left), we have two personalized subjects: a person named $<T>$ (subject 1) and a dog named $<bo>$ (subject 2). The personalized prompt would be: “$<T> is <token 1_1>...<token1_{16}>. <bo> is <token2_1>...<token2_{16}>.$”
>
> Examples with 3 personalized subjects can be found in the Rebuttal PDF file (Fig. 2).
>
> > **More discussion on the finetuning cost (e.g., wall clock time) in the rebuttal is appreciated.**
>
> The total time for learning a new subject with LLaVA-1.5-13B would be around 40 minutes.
> For a new subject with 3-5 images, the time required to create the training data is roughly 3 minutes (1-2 minutes for conversation creation and 1 minute for hard negative retrieval). The optimization process, including training new tokens, takes approximately 34 minutes for 15 epochs (measured on an A6000 machine).

---

> > ### Comment · Reviewer_ddkC · 2024-08-12
> > **Thank you for the rebuttal.**
> >
> > Thank the authors for the rebuttal. My confusions are positively cleared. I keep my positive rating.

---

### Official Review · Reviewer_VkWL · 2024-07-06

**Soundness:** 3
**Presentation:** 4
**Contribution:** 2
**Rating:** 5
**Confidence:** 5

**Summary:**

The paper attempts to personalize LLM's by adding personal details like dog etc. The overall idea is to add the corresponding tokens in the LLM and fine-tune the last output layer for the newly added tokens in the embedding space. This results in the ability to do personalized question answering and recognition.

**Strengths:**

1. The paper is nicely written with all the details covered with figures that improve the understanding.

2. The overall idea is good, and personalization is an important trait that current LMMs are missing.

3. The proposed approach is simple, and the model is easily able to learn new personalized concepts.

**Weaknesses:**

1. The overall paper relies on the fact that LLaVA-like architecture cannot do multi-image conversations (L253) and the only available model is GPT-V at present, which is not open source. I believe this is a strong assumption. Given the strong performance of GPT-V, it is also likely that a LLaVA-like open source model that supports multi-image conversation will achieve good performance when the image is given in the system prompt and then asked to recognize or do QA (similar to the GPT-V experimental setup).

2. The current architecture scheme does not look scalable. To add one concept, the model has to add (k+1) tokens. A person can have typically 10 personalized concepts that they might want to add, so the token space increased fast. Given the nature of the problem, it is essential to discuss what happens when there are multiple concepts that are to be learned.

3. (L307) It is true that GPT-V requires more tokens compared to your method, but the less token requirement is a direct consequence of training with the new token, whereas GPT-V is zero-shot. Hence, I do not find this paragraph appealing.

4. The recognition metrics is unclear to me. Given the four categories True Positive (TP), FP, FN, and TN, it is good to report F1 score and other scores that consider class imbalance. Is the average of positive and negative accuracy a standard metric?

5. (Fig. 4) The GPT-V graph keeps increasing with increase of prompting tokens. Is it possible to go one order more to see the saturation in the performance? The current trend makes it appear that it can easily go above Yo'LLaVA's performance.

**Questions:**

Please answer the weaknesses above.

W1 above requires justification on why the method is more than providing image context in the system prompt. It is understandable that current LLaVA architecture does not allow that. But, will this method still be useful when such an architecture is open sourced? Or provide justifications on why multi-image conversation model will not be released anytime soon.

Please answer other weaknesses as well that are crucial for the paper.

**Limitations:**

The authors have discussed the limitations and weaknesses and it correctly reflects the issues.

---

> ### Author Rebuttal · Authors · 2024-08-06
>
> > **The overall paper relies on the fact that LLaVA-like architecture cannot do multi-image conversations (L253) and the only available model is GPT-V at present**... [omitted]
>
> Yo’LLaVA can learn personalized subjects more efficiently using fewer tokens and more effectively encode visual attributes compared to strong prompting baselines (e.g., GPT-4V + Image prompting).
>
> It is evident that Yo’LLaVA, with just 17 tokens, outperforms GPT-4V with single image prompting (1k+ tokens) (Tab. 5). Notably, Yo’LLaVA, even with only 17 tokens, yields results almost comparable to GPT-4V using 5 images (5k+ tokens); see Fig. 4 (Lines 286-288).
> Unfortunately, GPT-4V is a closed-source model, so we cannot integrate Yo’LLaVA with GPT-4V to reduce the number of tokens used while retaining performance.
>
> Recently, Meta AI released Chameleon [1] - a LMM that supports multiple images/text as inputs and images/text as outputs. We have tested Yo'LLaVA with Chameleon, and it has demonstrated superior results in image generation! (See Rebuttal PDF, Fig. 1). These early results show that not only can Yo’LLaVA be adapted to other multimodal models (e.g., Chameleon), but it can also provide clear advantages!
>
> > **The current architecture scheme does not look scalable**... [omitted]
>
> Yo’LLaVA is much more token-efficient while achieving similar performance with Image Prompting.
>
> As noted by the reviewer (in above comment about Image prompting vs. Yo’LLaVA), consider a scenario where we need to personalize 10 subjects with two options: (1) Yo’LLaVA and (2) Image Prompting.
> * Using Yo’LLaVA: We would add 10 (subject) x (16+1) (token/subject) = 170 tokens
> * Using Image Prompting: We would add 1 (reference image) x 10 (subject) x 1024 (token/image) = 10,240 tokens (we use 1024 tokens here which is the number of tokens for an image used in Chameleon [1], since the number of tokens is unknown for GPT-4V).
> This clearly illustrates the significant token efficiency of Yo’LLaVA!
>
> > **less token requirement is a direct consequence of training with the new token, whereas GPT-V is zero-shot**... [omitted]
>
> As stated in the main paper, GPT-4V + Image Prompting baseline is meant to show accuracy and efficiency tradeoff of a SOTA proprietary multi-image understanding model with zero-shot inference time personalization (Line 257 “Since images convey more information than text, we hypothesize that personalized image prompts represent upper bound for prompting effectiveness.”).
>
> We agree that requiring fewer tokens is a direct result of training new tokens. However, the primary goal of this paper is to introduce a new problem setting: Personalizing LMMs. Our work is the first to demonstrate its feasibility: we proposed a new novel approach and showed promising results: Yo’LLaVA is much more token-efficient (17) while achieving similar performance with image prompting (5k~)!
>
> We anticipate that integrating Yo’LLaVA with GPT-4V could significantly reduce the number of tokens used while retaining performance; however, we could not try this since GPT-4V is a closed-source framework (Line 289-290).
>
> > **The recognition metrics is unclear to me**... [omitted]
>
> We have 40 subjects, each with 5-10 test images containing the corresponding subject.
> For each subject, all of its test images serve as positive test images, while test images from the remaining 39 categories serve as negative test images.
>
> During testing, we show a photo to the model and ask, “Can you see if <sks> is in this photo? Answer with a single word or phrase.” The ground-truth response is “Yes” for photos containing <sks>, and “No” for others. An extension of Tab. 5 with F1 score is:
>
> | | Ours | LLaVA | LLaVA + Prompt | GPT-4V + Prompt |  |
> |------------------|-------|-------|-------------------|--------------------|-------|
> | Type | Learnable | Text | Human | Text | Human | Image |
> | # tokens  | 16    | 0     | 16 | 1.3k | 1k    | 5k   |
> | F1 | 0.93  | 0.00  | 0.80 | 0.48 | 0.81  | 0.81 | 0.89 | 0.92 |
>
> As explained in Line 269-274, we purposely report weighted accuracy = 0.5 ∗ accuracy positive + 0.5 ∗ accuracy negative because:
> * In personalization setting, the ability to recognize the positive class (it is <sks>) and not-sks (it is not <sks>) is equally important
> * The positive/ negative class are heavily imbalanced
>
> We also report accuracy for each positive and negative class.
>
> > **It appear that GPT-4V can easily go above Yo'LLaVA's performance** ... [omitted]
>
> Our setting involves visual personalization, which typically requires 3-5 images per user, as initially proposed by Textual Inversion/ Dreambooth [2]. In our experiment, we provided 5 images/subject to GPT-4V, which is similar to the number used to train Yo’LLaVA. As shown in Tab. 5, we achieved comparable performance while using significantly fewer tokens (17 vs. 5k). Unfortunately, we cannot extend to an order of magnitude more images (e.g., ~10) as suggested by the reviewer, just to see if GPT-4V will surpass Yo’LLaVA in performance, this is because:
>
> -   As stated in Line 258, comparison to GPT-4V is solely for reference as it’s closed-source framework.
> -   With 5 images, GPT-4V already surpassed Yo’LLaVA in terms of recognition accuracy (0.925 vs. 0.924; Line 287); however, use significantly more tokens (5000+ vs. 17!)
> -   The best setting would be to directly integrate Yo’LLaVA with GPT-4V for both performance and token-efficiency – Unfortunately, we cannot do that as GPT-4V is closed-sourced! (Line 288).
>
> Also, there is no existing dataset to support this (e.g., dataset for personalization that have 10+ imgs/subject), and we believe this is not a practical setting. If the reviewer is aware of any such dataset, we would much appreciate the reference.
>
> *Reference:*
>
> *[1] Chameleon Team, Chameleon: Mixed-Modal Early-Fusion Foundation Models, arXiv, 2024*
>
> *[2] Ruiz et al, DreamBooth: Fine Tuning Text-to-Image Diffusion Models for Subject-Driven Generation, CVPR, 2023*

---

> > ### Comment · Reviewer_VkWL · 2024-08-08
> > **Requesting clarifications**
> >
> > Hi,
> >
> > Thanks for the rebuttal and I appreciate the response.
> >
> > While I am taking a deeper look at the responses, can you please explain this calculation a bit more?
> >
> > > Using Image Prompting: We would add 1 (reference image) x 10 (subject) x 1024 (token/image) = 10,240 tokens (we use 1024 tokens here which is the number of tokens for an image used in Chameleon [1], since the number of tokens is unknown for GPT-4V). This clearly illustrates the significant token efficiency of Yo’LLaVA!
> >
> > What I meant by image prompting is a conversation like this:
> >
> > ```
> > User: This is my cat whose name is <name> and here is how he/she looks: <add_image_tokens_or_embeddings>. Please respond to next questions personalized to my cat.
> > .......
> > User: I'm thinking about buying a birthday gift for <bo>. What do you recommend? (From your figure1).
> > ```
> >
> > How is this adding 1024 tokens to the model? I understand that the inference may use more tokens, but can you elaborate the calculation? Is it for training or during evaluation? Why does and in-context few-shot learning like this will not work for models with image understanding capabilities?
> >
> > Thanks again for your clarifications and responses.

---

> > > ### Author Response · Authors · 2024-08-10
> > >
> > > We apologize for the confusion.
> > > In the rebuttal, we had written, *“We would add… ”* referring to the inference time but a more precise description would be: *“We would add… **into the system prompt**.”*
> > >
> > > Here, we give more details on the calculation differences between: (1) Yo’LLaVA and (2) Image Prompting, across three aspects: (A) Approach, (B) Training, and (C) Inference.
> > >
> > > **A - Approach:**
> > >
> > > * Yo’LLaVA: Learns to embed visual details about the subject into a personalized learnable soft-prompt:
> > > "$<sks>$ is $<token_1>...<token_k>$".
> > > In the experiments, we choose $k=16$, which results in 17 tokens ($<sks>$, $<token_1>$, $<token_{16}>$) used for a personalized subject.
> > >
> > > * Image Prompting: Uses $n$ image(s) to represent the personalized subject:
> > > "Here is a photo(s) of $<sks><image_1>...<image_n>$".
> > > For simplicity of this computation, we use $n=1$ (In Table 5, we tried GPT-4V + Image Prompting with $n=1$ and $n=5$ images).
> > > In Chameleon [1], an image $<image_i>$ is represented by 1024 tokens; thus, in this case, 1024 tokens are used to represent the personalized subject. (We use Chameleon because the number of tokens used to represent an image in GPT-4V is unknown.)
> > >
> > > In both approaches, the personalized prompt is added to the system prompt.
> > >
> > > **B - Training**
> > >
> > > - Yo’LLaVA: learns 17 tokens (in which, token $<sks>$ is added into the vocabulary (Please refer to Line 152-157 for further details)), which takes roughly 40 minutes to train (measured with LLaVA-1.5-7B on an A6000 machine).
> > > - Image Prompting: no training needed.
> > >
> > > **C - Inference**
> > >
> > > C.1. Consider we only have one personalized subject called $<sks^1>$
> > >
> > > - Yo’LLaVA: Add "$<sks^1>$ is $<token^1_1>...<token^1_{16}>$" to the system prompt.
> > > In this case, the number of tokens used to represent a personalized subject is: 1 (subject) x 17 (tokens/subject) = **17 tokens**.
> > >
> > > - Image Prompting: Add "Here is a photo of $<sks^1><image^1_1>$”.
> > > In this case, the number of tokens used to represent a personalized subject with a single image $<image^1_1>$ is: 1 (subject) x 1 (reference image) x 1024 (tokens/image) = **1024 tokens**.
> > >
> > > C.2. Consider we have 10 personalized subjects $<sks^1>$, …, $<sks^{10}>$
> > > - Yo’LLaVA: Add to system prompt:
> > > “$<sks^1> is <token^1_1>...<token^1_{16}>$.
> > > $<sks^2> is <token^2_1>...<token^2_{16}>$.
> > > ...
> > > $<sks^{10}> is <token^{10}_1>...<token^{10}_{16}>$”.
> > > In this case, we use 10 (subject) x 17 (tokens/subject) = **170 tokens** to represent 10 personalized subjects.
> > >
> > > - Image Prompting: Add to system prompt
> > > "Here is a photo of $<sks^1><image^1_1>$.
> > > Here is a photo of $<sks^2><image^2_1>$
> > > ….
> > > Here is a photo of $<sks^{10}><image^{10}_1>$”.
> > > In this case, we use 1 (reference image/ subject) x 10 (subject) x 1024 (tokens/image) = **10240 tokens** to represent 10 personalized subjects!
> > >
> > > ---
> > >
> > > We agree that Image Prompting is a strong baseline, and indeed, Table 5 shows that GPT-4V + Image Prompting (5 images) yields competitive or better results than Yo’LLaVA in both Recognition Accuracy (0.925 vs. 0.924) and Text Question Answering (0.987 vs. 0.883). However, it uses significantly more tokens (~5k vs. 17)! We hypothesize that integrating Yo’LLaVA with GPT-4V could further improve performance while being more token-efficient. Unfortunately, we cannot do this because GPT-4V is a closed-source model.
> > >
> > > It’s also worth noting that in our early experiments, we empirically find that Yo’LLaVA can be integrated with other multimodal models (e.g., Chameleon) and offers advantages in image generation quality (e.g., the generated images are much closer to the personalized subject, as shown in the Rebuttal file, Fig. 1).
> > >
> > > ---
> > >
> > > We thank reviewer for bringing up this discussion. We will include this in our revision to enhance the understanding between Yo'LLaVA and Image Prompting.
> > > We are happy to discuss and clarify any further questions or doubts.
> > >
> > > *Reference:*
> > >
> > > *[1] Chameleon Team, Chameleon: Mixed-Modal Early-Fusion Foundation Models, arXiv, 2024*

---

> > > > ### Comment · Reviewer_VkWL · 2024-08-10
> > > > **Thanks for the clarification**
> > > >
> > > > I thank the authors for the clarifications and clear comparisons for training and inference.
> > > >
> > > > While the overall idea makes sense, and the 40 mins training time is particularly convincing. However, I feel the method is still not scalable with so many diverse concepts that we have in everyday life.
> > > >
> > > > Nevertheless, I feel this is an interesting research problem with its own application -- inference-time cost saving. I have raised the rating to **borderline accept** and would like to see the paper getting accepted. I would not raise further since it is clear that the method has a critical limitation in an increased training requirement when in-context learning is comparable. Furthermore, as the number of subjects increase, the performance drop of the LLM for non-subject queries is bound to decrease (even if the decrease is not evident at this stage with a few tokens).
> > > >
> > > > Please add the above discussion in the final version of the paper.

---

> > > > > ### Author Response · Authors · 2024-08-10
> > > > >
> > > > > We sincerely appreciate the reviewer's insightful comments and the increased rating. We will make sure to incorporate this discussion into the revision.

---

### Official Review · Reviewer_PZy9 · 2024-07-14

**Soundness:** 3
**Presentation:** 4
**Contribution:** 3
**Rating:** 5
**Confidence:** 4

**Summary:**

The paper introduces Yo'LLaVA, a novel approach to personalizing Large Multimodal Models (LMMs) to handle user-specific concepts and contexts. The proposed method embeds a personalized subject into a set of latent tokens given a handful of example images, enabling personalized textual and visual conversations. The paper includes qualitative and quantitative analyses demonstrating the efficiency and effectiveness of Yo'LLaVA in comparison to existing prompting baselines.

**Strengths:**

1. The task of personalizing LMMs is novel and addresses a significant gap in current LMM capabilities, which are primarily designed for generic tasks. The proposed method has wide-ranging applications, including personalized assistants in health, education, and entertainment.
2. The approach of using learnable prompts to embed personalized concepts is innovative and well-justified. The method ensures the retention of broad pre-trained knowledge, which is crucial for maintaining the overall functionality of the model while adding personalized capabilities.
3. The paper provides both qualitative and quantitative analyses to validate the effectiveness of Yo'LLaVA. The comparisons with strong prompting baselines (e.g., LLaVA) highlight the improvements in terms of efficiency and the ability to encode visual attributes.

**Weaknesses:**

1. While the paper provides promising results, the evaluation is somewhat limited to specific tasks. A broader evaluation across more diverse tasks and real-world scenarios would strengthen the claims.
2. The paper does not thoroughly address the scalability of the proposed method. How well does Yo'LLaVA handle a large number of personalized subjects or frequent updates to personalized concepts? Discussing and testing the scalability of the approach would be beneficial.
3. The method proposed in this paper effectively addresses the issue of personalizing LMMs and highlights its advantages over providing a language description for a given object. However, what are the benefits of this approach compared to directly providing image of the object to the LMM, and then using a CoT to first prompt LMM to give detailed descriptions of the subject, and bind these descriptions with the speical token like <bo> in the paper. Besides,  I have also tried the proposed personalized setting on GPT4-o, and it seems that GPT4-o can directly handle this problem.

**Questions:**

Although only limited number of images are required, the process of curating the conversationing data and retrieving hard negatives should be executed for each personalized subject, which incurs a lot of human efforts.

**Limitations:**

See weakness.

---

> ### Author Rebuttal · Authors · 2024-08-06
>
> > **While the paper provides promising results, the evaluation is somewhat limited to specific tasks. A broader evaluation across more diverse tasks and real-world scenarios would strengthen the claims.**
>
> We agree that a broader evaluation across more diverse tasks would strengthen the paper.
>
> However, at this moment, we are not aware of any benchmark/ datasets for this personalization yet (e.g., personalized datasets in healthcare, business).
> Thus, we include two main types of tasks: Recognition (Sec. 5.1) and Question and Answering (Sec. 5.2), which are the foundational tasks of personalization (similar to MyVLM [1], in which introduce a novel dataset with 29 concepts, while our dataset has 40 concepts).
>
> We have also made an effort to test the model with some examples of real-world scenarios (e.g., Fig. 1, suggesting a gift for a pet’s birthday; or Table 3, Left, writing a Facebook caption). More examples can be found in Appendix, Table 9-19.
>
> > **The paper does not thoroughly address the scalability of the proposed method. How well does Yo'LLaVA handle a large number of personalized subjects or frequent updates to personalized concepts? Discussing and testing the scalability of the approach would be beneficial.**
>
> In Table 3 (Right), we indeed show examples of handling 2 subjects at a time (a person named <T> and her dog named <bo>). In the Rebuttal PDF, we show additional qualitative results of handling 3 subjects (a person named <T>, her dog named <bo>, and her cat named <mam>). If users want to update the personalized subject, the re-train process would take roughly 40 minutes.
>
> A further study on scalability of Yo’LLaVA would heavily rely on the availability of a large scale dataset/ benchmark for personalization. At this moment, we are not aware of any such available dataset/ benchmark for this task yet.
> To collect and label this dataset would be challenging, as (1) it depends on privacy (e.g., as stated in Line 31 “user is only willing to share 4-5 images of a person named <A>”); and (2) multiple subjects should be related (e.g., different subjects appear in the same photo like <T> and her pets <bo> and <mam>).
> We anticipate that future research will address these challenges!
>
> > **The method proposed in this paper effectively addresses the issue of personalizing LMMs and highlights its advantages over providing a language description for a given object. However, what are the benefits of this approach compared to directly providing image of the object to the LMM, and then using a CoT to first prompt LMM to give detailed descriptions of the subject, and bind these descriptions with the speical token like <bo> in the paper. Besides, I have also tried the proposed personalized setting on GPT4-o, and it seems that GPT4-o can directly handle this problem.**
>
> Thanks for your suggestion to use Chain of Thought (CoT) to improve the detail of descriptions about a subject. In limited rebuttal time, we have quickly tested LLaVA + CoT for more detailed descriptions, and recognition accuracy increased roughly 5% (from 0.819 to 0.852 (Table 5)). While increasing, this is still behind Yo’LLaVA (0.924) and Image Prompt (0.901-0.925).
>
> This is expected as discrete text will always contain some ambiguity and may not fully capture the visual details (e.g., for a dog named <bo>, the coat color might be described as “yellow,” but visually, yellow can have many different shades). The strongest baseline is directly providing the image prompt. We indeed tried this in the main paper with GPT-4V, as GPT-4V is currently a leading proprietary multimodal model. The results shown in Table 5 demonstrate that Yo’LLaVA can achieve similar performance to GPT-4V with 5k tokens, which is much more computationally efficient!
>
> With text-only response models (e.g., LLaVA, GPT-4V), Yo’LLaVA can help to retain the recognition accuracy while using much less tokens (e.g., increasing from 0.822 vs 0.924).
> We anticipate that when shifting to another modality (e.g., image/text generation models like Chameleon [2]), Yo’LLaVA also helps with image generation quality.
> We provide early results from our experiments integrating Yo’LLaVA and Chameleon in the Rebuttal PDF (Fig. 1). As shown, the optimized prompt captures much more precise visual details about the subject compared to the plain text description (e.g., Yo’LLaVA + Chameleon captures more details about a dog named <bo>)!
>
> These early results show that not only Yo’LLaVA can be adapted to other multimodal models (e.g., Chameleon), it can also provide clear advantages with clear text personalized prompts.
>
> > **Although only limited number of images are required, the process of curating the conversationing data and retrieving hard negatives should be executed for each personalized subject, which incurs a lot of human efforts.**
>
> All the training data is generated automatically, so no manual labeling is needed!
>
> For the conversation training data, we employ LLaVA to generate answers for each template question (Lines 215-216). The template questions are fixed and universal for all subjects (a complete list of these questions can be found in Appendix G, Tables 18-19). For the negative images, for each positive image, we retrieve the top m images with the highest CLIP image embedding similarity (Lines 191-192). This process is fully automated and does not involve any human supervision!
>
> *Reference:*
>
> *[1] Alaluf et. al., MyVLM: Personalizing VLMs for User-Specific Queries, ECCV, 2024*
>
> *[2] Chameleon Team, Chameleon: Mixed-Modal Early-Fusion Foundation Models, arXiv, 2024*

---

### Author Rebuttal · Authors · 2024-08-05

We introduce the *novel task of personalizing LMMs* and present *Yo'LLaVA* -- a framework to embed personalized subjects (e.g., your pet) into a comprehensible prompt for LLaVA.

We are encouraged by positive feedback from reviewers on our paper!
- **Originality**: “novel” (#PZy9, #wZcX), “good” (#VkWL), “important and interesting” (#ddkC)
- **Significance**: “addresses a significant gap in current LMM capabilities” (#PZy9), “an important trait that currently LMMs are missing” (#VkWL), “a valid first attempt to this task” (#ddkC).
- **Technicality**: “innovative and well-justified” (#PZy9), “easily able to learn new personalized concepts” (#VkWL), “makes full sense” (#ddkC)
- **Clarity**: All reviewers think the paper presentation is *“excellent”*!

---

We thank reviewers for their time and effort in reviewing our paper.
Answers to individual reviewers are addressed below each review.
Please let us know if you have any additional questions or concerns!

---

### Comment · Area_Chair_7Hdh · 2024-08-12
**Please read the author rebuttal, other reviews and respond to the authors NOW!**

Dear Reviewers,

Thanks to those of you who already responded to the authors acknowledging the rebuttal and asking follow-up questions if any.

Those who have not responded yet, please do the following NOW: thoroughly read the rebuttal, the other reviews and respond to the authors about whether all your questions / concerns have been addressed or not. If not, please elaborate on which questions / concerns are still not addressed so that the authors have fair chance of addressing them before the author-reviewer discussion period ends in ~41 hours from now (August 13th, 11:59pm AoE).

Your AC

---

### Decision · Program_Chairs · 2024-09-25

**Decision:**

Accept (poster)

**Comment:**

The reviewers find the problem of personalization of VLMs being studied in this paper to be relevant and important, the proposed method to be innovative, simple and well justified, and the experimental results to be strong. The reviewers had raised some concerns, but the rebuttal successfully addressed most of them and all reviewers recommend acceptance. The authors are recommended to improve the final paper version by following the reviewer recommendations.